# Phospholipid scramblase 1 (PLSCR1) regulates interferon-lambda receptor 1 (IFN-λR1) and IFN-λ signaling in influenza A virus (IAV) infection

Alina Xiaoyu Yang[1], Lisa Ramos-Rodriguez[1], Parand Sorkhdini[1], Dongqin Yang[1], Carmelissa Norbrun[1], Sonoor Majid[1], Sanghyun Lee[1], Yong Zhang[2], Michael Holtzman[2], David F Boyd[3], Yang Zhou[1]*

[1]Department of Molecular Microbiology and Immunology, Brown University, Providence, United States; [2]Division of Pulmonary and Critical Care Medicine, Washington University School of Medicine in St. Louis, St. Louis, United States; [3]Department of Molecular, Cell and Developmental Biology, University of California Santa Cruz, Santa Cruz, United States

*For correspondence:
yang_zhou@brown.edu

Competing interest: The authors declare that no competing interests exist.

## eLife Assessment

This **valuable** manuscript presents a potentially novel mechanism by which the phospholipid scramblase, PLSCR1, defends against influenza A virus infection. The strength of the paper rests on **solid** findings involving knockout and lung specific over-expressing Plscr1 mice, airway tissue expression and mechanistic studies to show Plscr1 enhances type III interferon-mediated viral clearance.

**Abstract** Phospholipid scramblase 1 (PLSCR1) is an interferon-stimulated gene (ISG) that has several known anti-influenza functions. However, the mechanisms in relation to its expression compartment and enzymatic activity have not been completely explored. Moreover, only limited animal models have been studied to delineate its role at the tissue level in influenza infections. Our results showed that influenza A virus (IAV)-infected *Plscr1*[-/-] mice exhibited exacerbated body weight loss, decreased survival rates, heightened viral replication, and increased lung damage. Interestingly, transcriptomic analyses demonstrated that Plscr1 was required for the expression of type 3 interferon receptor (Ifn-λr1) upon IAV infection by binding to its promoter. In addition, PLSCR1 interacted with IFN-λR1 on the cell surface of pulmonary epithelial cells following IAV infection, suggesting it also modulated IFN-λ signaling via protein-protein interactions. The lipid scramblase activity of PLSCR1 was found to be dispensable for its anti-flu activity. Finally, single-cell RNA sequencing data indicated that *Plscr1* expression was significantly upregulated in ciliated airway epithelial cells in mice following IAV infection. Consistently, *Plscr1*[floxStop];*Foxj1-Cre*[+]mice with ciliated epithelial cell-specific Plscr1 overexpression showed reduced susceptibility to IAV infection, less inflammation, and enhanced Ifn-λr1 expression, suggesting that Plscr1 primarily regulates type 3 IFN signaling as a cell-intrinsic defense factor against IAV in ciliated airway epithelial cells. Our research will elucidate virus-host interactions and pave the way for the development of novel anti-influenza drugs that target human elements like PLSCR1, thereby mitigating the emergence of drug-resistant IAV strains.

## Introduction

Influenza A virus (IAV) is highly contagious and causes acute respiratory infectious disease transmitted by virus-containing droplets. Its segmented gene and wide host range facilitate frequent antigenic shift during coinfection, posing a significant threat to human health. IAV has caused several pandemics in human history, and seasonal flu remains a major health burden in present days, despite annual vaccination efforts (*Holbach, 2016*). Existing anti-flu drugs mainly target influenza virus proteins: (1) penetration and shelling inhibitors, such as Gocovri (amantadine) and Flumadine (Rimantadine); (2) neuraminidase inhibitors, such as Relenza (zanamivir), Tamiflu (oseltamivir phosphate), Rapivab (peramivir), and Xofluza (baloxavir marboxil) (*The Medical Letter on Drugs and Therapeutics, 2020*). However, the emergence of drug-resistant variant strains is common. Moreover, these direct antivirals have a short therapeutic window and are the most effective only if given within the first 48 hr after the initial infection (*The Medical Letter on Drugs and Therapeutics, 2020*). In addition, IAV-induced acute inflammation could persist, leading to severe complications, such as life-threatening pneumonia, immunopathology, and acute respiratory distress syndrome (ARDS). Therefore, there is a pressing need to understand host immune responses and develop anti-influenza drugs targeting host factors.

At the center of anti-flu immunity are the interferon (IFN) pathways. Discovered a mere two decades ago, type 3 IFNs were initially perceived as redundant to type 1 IFNs, given their shared intracellular signaling pathways and antiviral activities. However, recent studies have revealed their unique properties, notably their signaling through a pair of heterodimer receptors (IFN-$\lambda$ R1/IL-10R2) distinct from type 1 IFN receptor complex (IFN-αR1/IFN-αR2) (*Sheppard et al., 2003*; *Donnelly et al., 2004*). First of all, IFN-$\lambda$ exhibits a more constrained expression pattern compared to type 1 IFNs. While the cellular sources of type 1 IFNs in viral infections depend on the infection route and the tissue tropism, they can be produced by a large variety of cell types, including epithelial, parenchymal, immune, and stromal cells, to combat infections in the skin, mucosal, organ, lymph node, and at the systemic level (*Swiecki and Colonna, 2011*). In contrast, IFN-$\lambda$ is primarily secreted by cells at barrier surfaces, such as respiratory and gastrointestinal epithelial cells, DCs, and macrophages (*Coccia et al., 2004*; *Sommereyns et al., 2008*). Moreover, their receptor distributions are vastly different. IFN-αR1/IFN-αR2 complex is present on nearly all cell types, but IFN-$\lambda$ R1/IL-10R2 complex is expressed exclusively on epithelial cells and a few immune cells, including neutrophils and subsets of dendritic cells (*Broggi et al., 2017*; *Yin et al., 2012*). Most importantly, IFN-$\lambda$ is produced earlier than type 1 interferons in IAV infection and can elicit effective antiviral responses without inflammation, such as the release of tissue-damaging mediators, such as tumor necrosis factor (TNF) and IL-1β from neutrophils. In fact, only under a high dose of IAV are type 1 interferons detected, contributing to tissue immunopathology (*Galani et al., 2017*). Another exclusive anti-flu mechanism of IFN-$\lambda$ is its role in preventing viral transmission from the upper airways to the lungs (*Klinkhammer et al., 2018*). Taken together, IFN-$\lambda$ provides a non-redundant front-line shield against influenza virus.

IFNs secreted by infected cells signal neighboring cells to enter an antiviral state by inducing the expression of hundreds of ISGs (*Chen et al., 2018*). PLSCR1, the most studied member of the phospholipid scramblase protein family, is one such ISG, with its expression highly induced by type 1, 2, and 3 interferons in various viral infections (*Zhou et al., 2000*; *Lu et al., 2007*; *Xu et al., 2023*). It is a type II transmembrane protein generally located on the cell membrane, but can be imported into the nucleus and act as a transcriptional factor to regulate several gene expressions by directly binding to their promoter regions (*Zhou et al., 2005*; *Huang et al., 2020*). PLSCR1 is detected in all tissues and has a relatively high expression in the lung. Although it is universally expressed across various cell types in the lung, including alveolar epithelial cells and lymphocytes, macrophages have the highest expression, followed by endothelial cells and respiratory ciliated cells (*Karlsson et al., 2021*). The main function of PLSCR1 is to catalyze Ca$^{2+}$-dependent, ATP-independent, bidirectional, and non-specific translocation of phospholipids between inner & outer leaflet of plasma membrane. Scrambling of membrane phospholipids results in the externalization of phosphatidylserine (PS), which acts as a docking site for many biological processes, including coagulation, apoptosis, and activation (*Bassé et al., 1996*).

Previous studies have described some critical anti-influenza activities of PLSCR1. For example, PLSCR1 interacts with the nucleoprotein (NP) of IAV, impairing its nuclear import and thereby suppressing virus replication in A549 cells (*Luo et al., 2018*). In addition, Plscr1 competes with

immunoglobulin-like domain-containing receptor 1 (ILDR1) for NP binding, inhibiting swine influenza virus (SIV) infection in mice (*Liu et al., 2022*). Besides direct interactions with the virus, PLSCR1 interacts with toll-like receptor (TLR) 9 and regulates its trafficking and ability to induce type I interferon production in plasmacytoid dendritic cells (*Talukder et al., 2012*). Furthermore, PLSCR1 is required to potentiate the expression of numerous other ISGs in response to IFN-β in Hey1B cells, including p56, ISG15, and OAS (*Dong et al., 2004*). The goals of this project are to determine the roles of Plscr1 in an IAV-infected mouse model, to implicate its involvement in IFN-$\lambda$ signaling, and to elucidate the cell types responsible for Plscr1-mediated anti-influenza activities.

To address this knowledge gap, we have employed a systematic approach, leveraging both global and cell type-specific *Plscr1*$^{-/-}$ and *Plscr1* knock-in overexpression mice infected with mouse-adapted human IAV. Human respiratory epithelial cell lines harboring PLSCR1 mutations have also been used to elucidate the subcellular roles of PLSCR1. Our findings demonstrate that Plscr1 is an IFN-$\lambda$-stimulated gene, and it protects mice against IAV infection by regulating *IFNLR1* gene expression in the nucleus and interacting with IFN-$\lambda$R1 protein on the cell membrane in ciliated airway epithelial cells. Our research will aid in the understanding of virus-host interactions and the development of novel anti-influenza therapeutics targeting human components, preventing the emergence of drug-resistant IAV strains.

## Results

### Plscr1 suppresses viral replication and protects mice in IAV infection

To uncover any anti-influenza functions of Plscr1 in mice, we employed WT and *Plscr1*$^{-/-}$ mice and exposed them to IAV infection. We pursued a systematic approach, including multiple endpoints and parameters to monitor mouse health, viral replication and dissemination, antiviral and inflammatory responses, and tissue damage (*Figure 1A*). We established 300 plaque-forming units (pfu) of IAV as the sublethal dose, and 900 pfu as the lethal dose, resulting in ~50% mortality in WT mice (LD50). Significantly higher *Plscr1* expression was observed in WT infected mice at 3 days post-infection (dpi) (*Figure 1B*), suggesting that Plscr1 is induced and potentially functions as an antiviral ISG in IAV infection in vivo.

Weight loss generally started at 3 dpi for both mouse strains. WT infected mice reached their lowest weights around 8 dpi, while *Plscr1*$^{-/-}$ infected mice experienced continued weight loss for an additional 2 days. *Plscr1*$^{-/-}$ mice exhibited significantly greater weight loss compared to WT mice with both sublethal and lethal dose infection at multiple time points (*Figure 1C and D*). Moreover, upon infection with a 900 pfu lethal dose of IAV, *Plscr1*$^{-/-}$ mice had a significantly lower survival rate (*Figure 1E*). In fact, 100% of *Plscr1*$^{-/-}$ mice died with 900 pfu of IAV in our experiment by 10 dpi.

We used three assays to robustly and comprehensively measure the viral burden in these animals. First, qRT-PCR analysis using cryopreserved lungs showed that IAV M gene segment mRNA was significantly higher in *Plscr1*$^{-/-}$ lungs at 3 dpi (*Figure 1F*). This observation was further supported by the plaque assay, which measures infectious virus particles, and the IAV titer was significantly higher in *Plscr1*$^{-/-}$ lungs at 3 dpi (*Figure 1G*). Finally, to visualize IAV burden and spread from upper respiratory tract, paraffin-embedded lung sections were stained for H1N1-FITC. In contrast to WT lungs where IAV infection was mostly local and restricted to major airways. *Plscr1*$^{-/-}$ lungs had viral dissemination extending from bronchioles to alveoli, with ~50% of lungs infected. Moreover, *Plscr1*$^{-/-}$ lungs had significantly higher corrected total cell fluorescence (CTCF) than WT lungs at 3 dpi, suggesting an overall higher viral burden during acute infection in the absence of Plscr1 (*Figure 1H*).

### Plscr1 limits innate immunity-mediated inflammation and lung damage in IAV infection

Bronchoalveolar lavage (BAL) was harvested for inflammatory cell counts using Cytospin followed by Diff-Quik stain. Immune cells infiltrated the lungs starting at 3 dpi and persisted at least until 10 dpi. *Plscr1*$^{-/-}$ mice had significantly higher total BAL cell counts at 7 dpi compared to WT mice, indicating a heightened inflammatory environment in alveoli during acute infection in the absence of Plscr1 (*Figure 2A*). IAV attracted neutrophils in early infection (3–7 dpi) and lymphocytes in late infection (7–10 dpi). Interestingly, *Plscr1*$^{-/-}$ lungs were significantly more neutrophilic at 3 dpi, and neutrophil populations persisted until 10 dpi (*Figure 2B*).

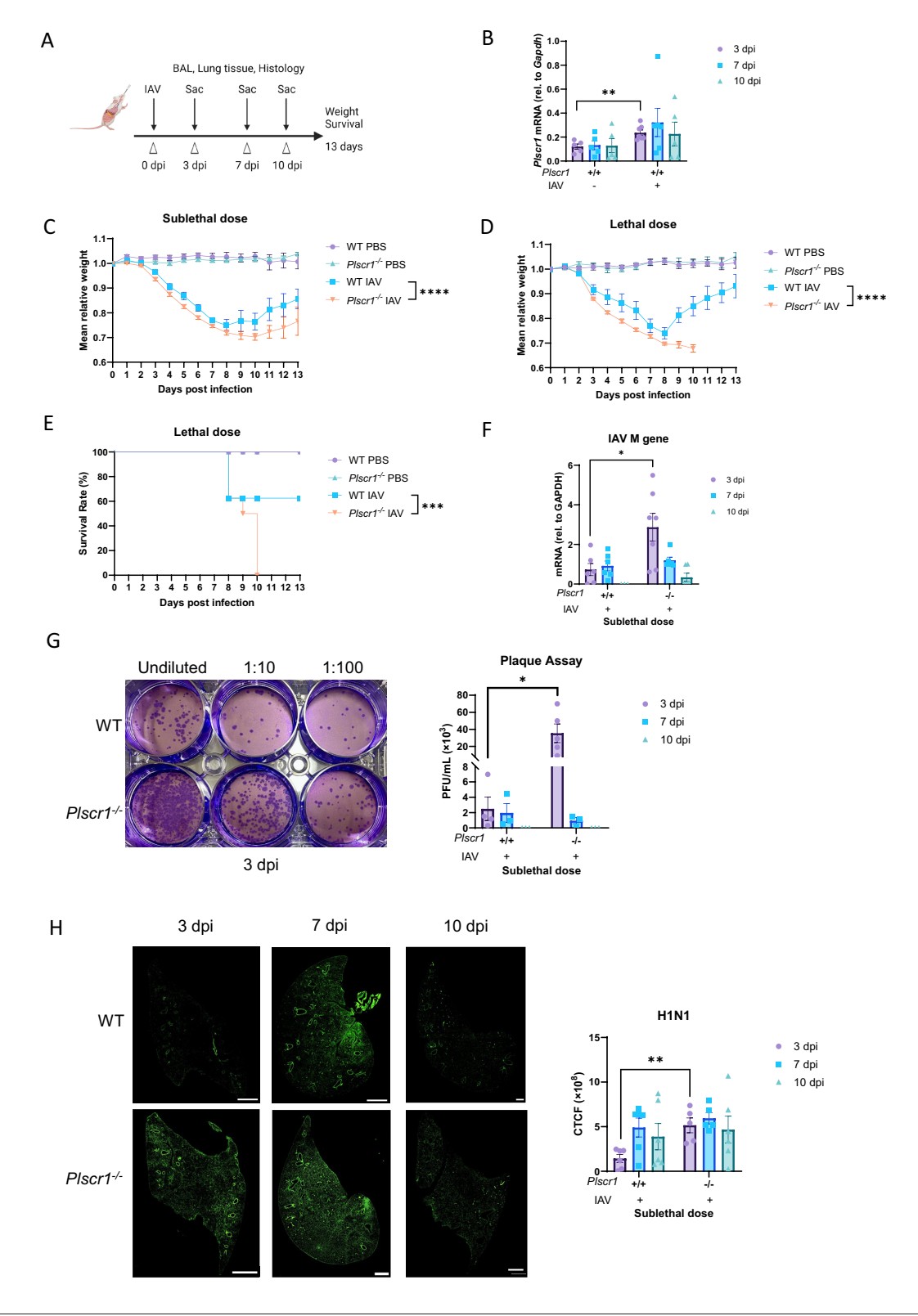

**Figure 1.** Increased susceptibility of *Plscr1⁻/⁻* mice to influenza virus infection. Wild-type (WT) and *Plscr1⁻/⁻* mice were exposed to sublethal (300 pfu, **B, C, and F–H**) or lethal (900 pfu, **D and E**) influenza A virus (IAV) (WSN) infection. (**A**) Scheme of experiment. (**B**) Whole lungs of WT mice were analyzed for *Plscr1* RNA by qRT-PCR. (**C, D**) Mean relative weight of mice post-sublethal or lethal infection. (**E**) Survival rate of mice post-lethal IAV infection. (**F**) Viral RNA load in the lungs was assessed by quantifying M gene by qRT-PCR. (**G**) Infectious viral titer in the lungs was assessed by plaque assays.

*Figure 1 continued on next page*

*Figure 1 continued*

(**H**) Representative staining for H1N1 in lungs. The scale bars represent 1 mm. Quantification was performed using ImageJ. Data are expressed as mean ± SEM of n=30 mice/group for weight loss post-sublethal infection and n=8 mice/group for weight loss and survival rate post-lethal infection. For the rest analysis, n=5–10 mice/group. All data were pooled from three independent experiments and described biological replicates. Log-rank (Mantel-Cox) test was used to compare survival rates. Ordinary two-way ANOVA tests were used to compare weight losses. *$p<0.05$, **$p<0.01$, ***$p<0.001$, ****$P<0.0001$. dpi, days post-infection. CTCF, Corrected Total Cell Fluorescence.

Lung sections were stained with Hematoxylin and Eosin (H&E) and histopathology was assessed. *Plscr1⁻/⁻* lungs showed minor and localized tissue damage as early as at 3 dpi, when WT lungs appeared normal. Consistently, aggravated immunopathology was evident in *Plscr1⁻/⁻* lungs at 7 and 10 dpi compared to WT lungs (*Figure 2C*). These observations included increased thickening and collapse of alveolar walls, pulmonary edema surrounding alveolar walls, inflammatory cell infiltration in peri-bronchial and parenchymal areas, and hyperemia in the absence of Plscr1, suggesting its role in maintaining tissue homeostasis in IAV infection.

To assess the role of Plscr1 as an ISG, IFN expressions in whole lungs were measured by qRT-PCR. We found significantly elevated levels of *Ifna*, *Ifnb*, *Ifng*, and *Ifnl* expression in *Plscr1⁻/⁻* mice, particularly in early infection (*Figure 2D*). This upregulation was associated with heightened production of Tnf-α and Ifn-λ in the BAL at 3 dpi, as assessed by ELISA (*Figure 2E*). These cytokines may implicate in hyperactive feedforward inflammatory circuits during early IAV infection leading to acute lung injury in *Plscr1⁻/⁻* mice (*Brandes et al., 2013*), consistent with worsened histopathology. On the contrary, the expressions of these mediators in WT mice were well controlled, facilitating virus elimination without inciting excessive inflammation. These findings underscore the antiviral and potentially immunoregulatory role of Plscr1.

## Plscr1 binds to *Ifnlr1* promoter and activates *Ifnlr1* transcription in IAV infection

To determine if IFN signaling pathways are regulated by Plscr1 during IAV infection, RNA sequencing of a total of 20,700 genes was performed using pooled samples from WT and *Plscr1⁻/⁻* mouse lungs infected with 300 pfu of IAV. A comprehensive examination of interferons and their receptors revealed that *Ifnlr1* expression was significantly lower in *Plscr1⁻/⁻* mice, despite high expression of *Ifnl* at both 3 and 7 dpi (*Figure 3A*). Importantly, *Ifnlr1* was the only IFN receptor exhibiting this pattern, as the expression of its coreceptor *Il10rb* and other IFN receptors remained unaltered. Inflammatory cytokines, including type 1 and 2 IFNs, *Tnf*, and *Il1b* were also highly expressed in *Plscr1⁻/⁻* infected mice, suggesting upstream cytokine production pathways were intact. The RNA sequencing result was further validated by qRT-PCR, which showed that *Plscr1⁻/⁻* mice failed to upregulate *Ifnlr1* expression at both 3 and 7 dpi compared to WT mice (*Figure 3B*). Furthermore, using the online Interferome database (*Rusinova et al., 2013*), we identified a total of 1113 ISGs in our dataset with a fold change ≥2 (*Figure 3C*). Enlarged heatmaps with gene names are provided in *Figure 3—figure supplements 1–3*. Among those ISGs, 584 are regulated exclusively by type 1 IFNs, and 488 are regulated by both type 1 and type 2 interferons. Unfortunately, the Interferome database does not include information on type 3 IFN-inducible genes in mice. Although many ISGs were robustly upregulated in *Plscr1⁻/⁻* infected lungs, consistent with inflammation data, a large subset of ISGs failed to be transcribed when *Ifnlr1* function was impaired, especially at 7 dpi. We suspect that those non-transcribed ISGs in *Plscr1⁻/⁻* mice may be specifically regulated by type 3 IFN and represent interesting targets for future research. As a result, disrupted type 3 interferon signaling due to impaired expression of *Ifnlr1* may underlie the delayed viral clearance and exacerbated immunopathology observed in *Plscr1⁻/⁻* mice (*Galani et al., 2017*).

To corroborate the cell-specific localization of Ifn-λr1 expression, WT IAV-infected mouse lungs were probed with immunofluorescent antibodies to detect Ifn-λr1 and Foxj1 (a marker for ciliated epithelial cells), Uteroglobin (a marker for club cells), or Sftpc (a marker for type 2 alveolar epithelial (AT2) cells). Consistent with previous studies, Ifn-λr1 was present on all of these epithelial cell types (*Figure 3D*). We further quantified Ifn-λr1 expression in selected airways by measuring mean intensity to disregard the differences in airway areas. Ifn-λr1 expression in alveoli were measured using Corrected Total Cell Fluorescence (CTCF), so that all fluorescent signals within the same size of alveolar areas were analyzed. Expression of Ifn-λr1 was significantly reduced in *Plscr1⁻/⁻* mouse

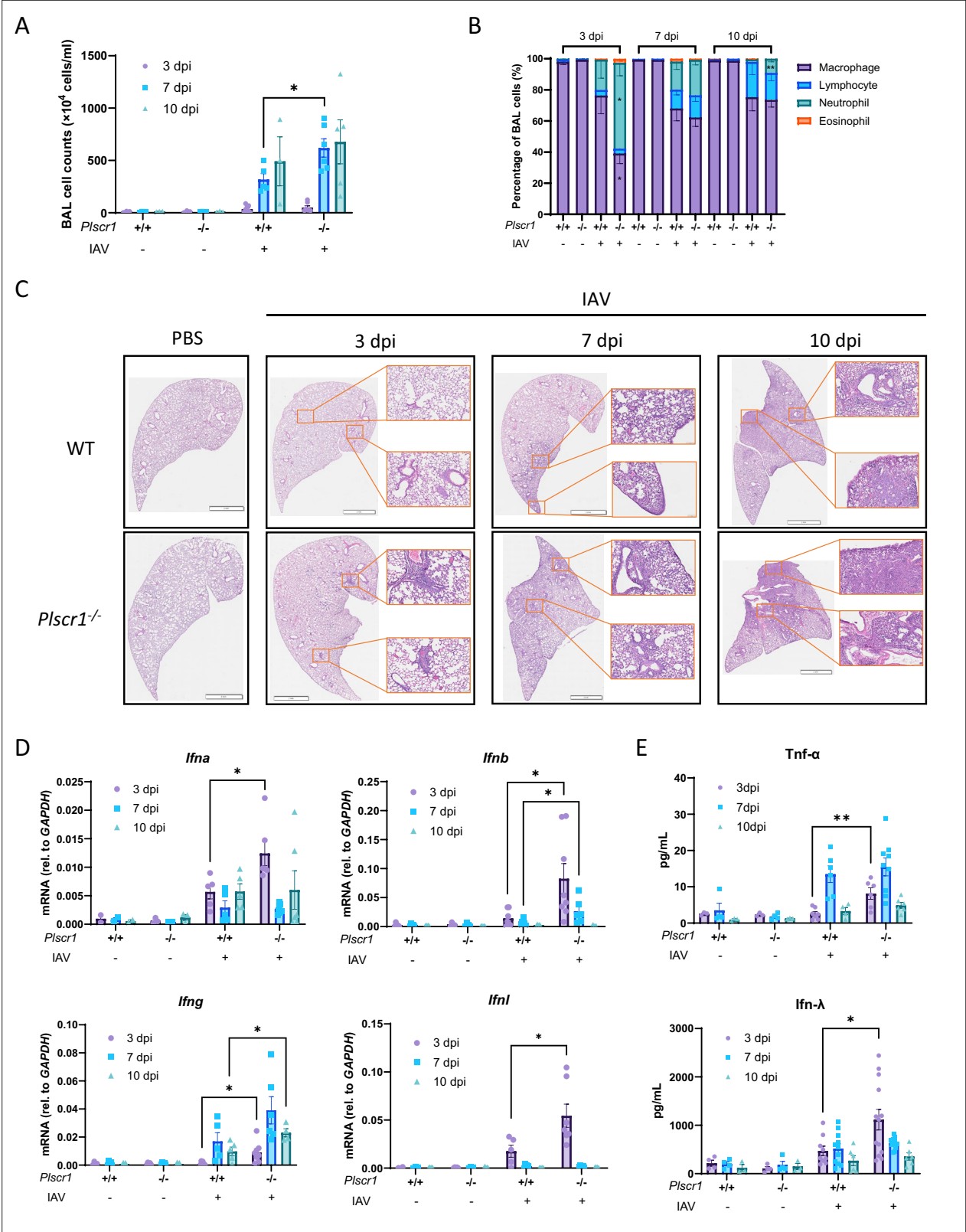

**Figure 2.** Increased lung inflammation in *Plscr1⁻ᐟ⁻* mice in influenza virus infection. Wild-type (WT) and *Plscr1⁻ᐟ⁻* mice were exposed to sublethal (300 pfu) influenza A virus (IAV) (WSN) infection. (**A**) Total Bronchoalveolar lavage (BAL) leukocyte numbers. (**B**) Differential cell counts in BAL. (**C**) Representative lung sections stained with Hematoxylin and Eosin (H&E). Scale bars represent 3 mm (main) and 200 μm (inlays). (**D**) Whole lungs were analyzed for *Ifna*, *Ifnb*, *Ifng,* and *Ifnl* RNA by qRT-PCR. (**E**) Tnf-α and Ifn- λ concentrations in BAL by ELISA. Data are expressed as mean ± SEM of n=3–14 mice/group. All data were pooled from three independent experiments and described biological replicates. *$p<0.05$, **$p<0.01$. dpi, days post-infection.

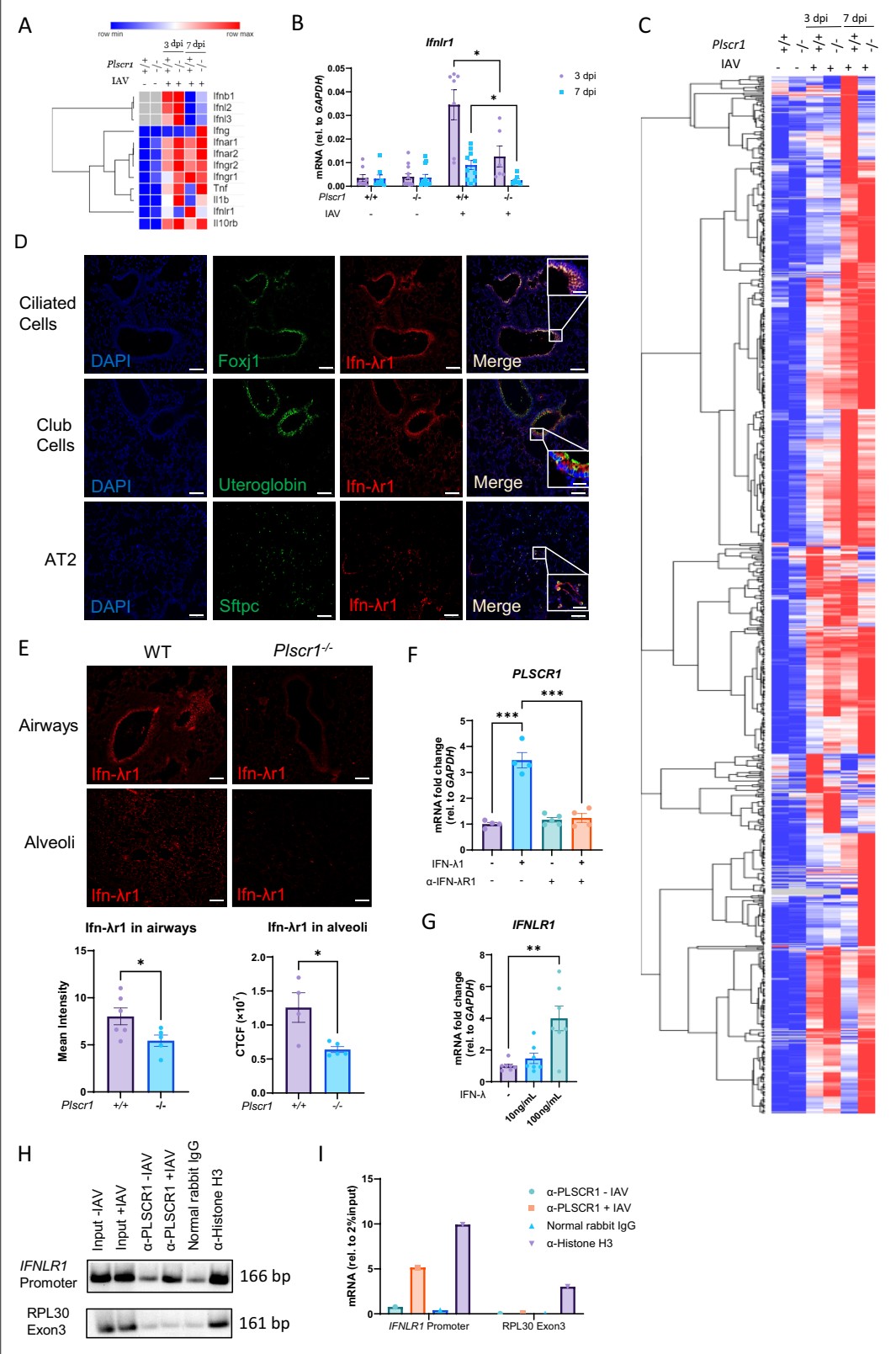

**Figure 3.** Transcriptional regulation of *IFNLR1* by *PLSCR1* and IFN-λ in influenza A virus (IAV) infection. (**A–E**) Wild-type (WT) and *Plscr1⁻ᐟ⁻* mice were exposed to sublethal (300 pfu) IAV (WSN) infection. (**A**) Heatmap of interferons and their receptors in whole lungs by RNA-seq. (**B**) Whole lungs were analyzed for *Ifnlr1* by qRT-PCR. (**C**) Heatmap of differential expressions of all interferon-stimulated genes (ISGs) in whole lungs by RNA-seq.

*Figure 3 continued on next page*

*Figure 3 continued*

Gene expressions were compared between groups within each row and color-labeled from row minimum (blue) to row maximum (red). (**D**) Localization of Ifn-$\lambda$r1+ cells in the lungs of IAV-infected WT mice at 7 dpi. Sections stained for Ifn-$\lambda$r1 (red), Foxj1, uteroglobin, or Sftpc (green), and DAPI (blue) are shown. Scale bars represent 50 μm (main) and 20 μm (inlays). (**E**) Representative staining for Ifn-$\lambda$r1 in airways or alveoli of IAV-infected WT and *Plscr1*$^{-/-}$ mice at 7 dpi. Scale bars represent 50 μm. Quantifications were performed using ImageJ. (**F, G**) Calu-3 cells were analyzed for *PLSCR1* (**F**) and *IFNLR1* (**G**) RNA by qRT-PCR after recombinant IFN-$\lambda$ and/or $\alpha$-IFN-$\lambda$R1 antibody treatment. Data are presented as fold change compared to non-treated group. (**H, I**) Chromatin-Immunoprecipitation of PLSCR1 and *IFNLR1* promoter in Calu-3 cells followed by standard PCR (**H**) and real-time quantitative PCR (**I**). Data are expressed as mean ± SEM of n=4–12 mice or wells/group. For transcriptomic analysis, 9 mice from each PBS-treated group and 4 mice from each IAV-infected group were pooled together. All data were pooled from three independent experiments and described biological replicates. \*$p<0.05$, \*\*$p<0.01$, \*\*\*$p<0.001$. dpi, days post-infection. CTCF, Corrected Total Cell Fluorescence. Scale bars represent 50 μm.

The online version of this article includes the following source data and figure supplement(s) for figure 3:

**Source data 1.** PDF file containing original gel for *Figure 3H*, indicating the relevant treatments.

**Source data 2.** Original gel corresponding to *Figure 3H*.

**Figure supplement 1.** Heatmap of differential expressions of all interferon-stimulated genes (ISGs) in whole lungs by RNA-seq (part 1), (part 2), (part 3).

**Figure supplement 2.** Heatmap of differential expressions of all interferon-stimulated genes (ISGs) in whole lungs by RNA-seq (part 1), (part 2), (part 3).

**Figure supplement 3.** Heatmap of differential expressions of all interferon-stimulated genes (ISGs) in whole lungs by RNA-seq (part 1), (part 2), (part 3).

**Figure supplement 4.** Requirement of Plscr1 in IFN-$\lambda$ signaling Iindependent of viral titer.

---

lungs, in both airways and alveoli regions (*Figure 3E*). Human Calu-3 epithelial cells were then used to investigate if *PLSCR1* expression can be regulated by IFN-$\lambda$. Using qRT-PCR, we found *PLSCR1* transcription was significantly increased when stimulated with IFN-$\lambda$, and this effect was attenuated by pre-incubation with $\alpha$-hIFNLR1 neutralizing antibody (*Figure 3F*). Additionally, IFN-$\lambda$ could directly stimulate the expression of its receptor, *IFNLR1*, thereby establishing a positive feedback loop to further drive the expression of *PLSCR1* (*Figure 3G*). These studies demonstrate that *PLSCR1* is an ISG that can be directly stimulated by IFN-$\lambda$ in airway epithelial cells.

We subsequently investigated the mechanism underlying Plscr1's transcriptional regulation of *Ifnlr1* in airway epithelial cells. Chromatin immunoprecipitation (ChIP) followed by standard PCR in IAV-infected Calu-3 cells unveiled that Plscr1 physically bound to the promoter region of *Ifnlr1*, with this binding becoming more evident in IAV-infected cells (*Figure 3H*). These results were further validated using real-time quantitative PCR, suggesting that Plscr1 translocated into the nucleus of lung epithelial cells upon IAV infection to activate *Ifnlr1* transcription (*Figure 3I*).

To further determine the regulation of Ifn-$\lambda$r1 by Plscr1 in response to Ifn-$\lambda$ signaling without the complexities associated with live virus infection, high molecular weight poly(I:C) was administered intranasally to mice daily for 6 days at 2.5 mg/kg of body weight, and mice were sacrificed on the following day (*Figure 3—figure supplement 4A*). We observed an increase in total BAL cell counts with poly(I:C) administration, although there was no difference between WT and *Plscr1*$^{-/-}$ mice (*Figure 3—figure supplement 4B*). However, *Plscr1*$^{-/-}$ mice exhibited a significantly higher number of lymphocytes in BAL (*Figure 3—figure supplement 4C*), indicating that repetitive poly(I:C) administration might activate adaptive immune responses rather than the innate immune system in an acute IAV infection (*Figure 2B*). All interferon expressions were elevated following poly(I:C) administration, with comparable levels observed between WT and *Plscr1*$^{-/-}$ mice (*Figure 3—figure supplement 4D*). Consistently, lung histopathology exhibited similar levels of inflammation or tissue damage in both WT and *Plscr1*$^{-/-}$ mice (*Figure 3—figure supplement 4E*). Importantly, *Plscr1* expression was significantly induced by poly(I:C) (*Figure 3—figure supplement 4F*). Notably, although no other phenotypical or genotypical variances were observed in poly(I:C)-treated *Plscr1*$^{-/-}$ mice, *Ifnlr1* expression was significantly lower in these mice, further affirming the requirement of Plscr1 for *Ifnlr1* expression in response to IFN-$\lambda$ (*Figure 3—figure supplement 4G*).

# Plscr1 interacts with Ifn-λr1 on pulmonary epithelial cell membrane in IAV infection

We employed a combination of multiple biophysical approaches to fully assess the potential protein interaction between Plscr1 and Ifn-λr1. Co-immunoprecipitation (Co-IP) followed by western blot

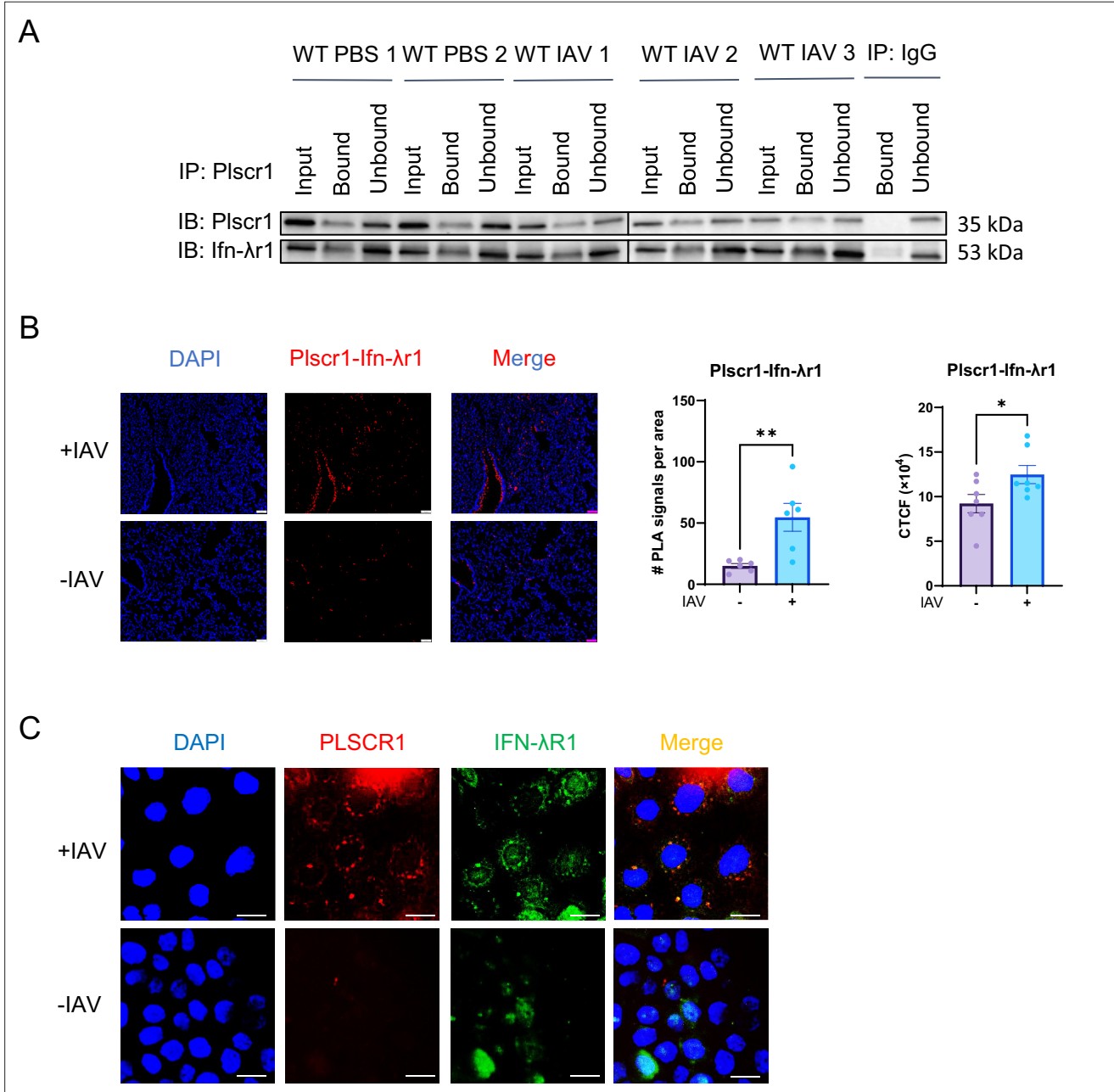

**Figure 4.** Protein interaction between IFN-λR1 and PLSCR1 in influenza A virus (IAV) Infection. (**A**) Co-Immunoprecipitation of Plscr1 and Ifn-λr1 in whole mouse lungs followed by western blot. (**B**) Proximity ligation assay of Ifn-λr1 and Plscr1 in the lungs of wild-type (WT) mice infected or uninfected with IAV. Scale bars represent 50 µm. Quantifications were performed using ImageJ. (**C**) Colocalization of IFN-λR1 (green) and PLSCR1 (red) on Calu-3 cell membranes infected or uninfected with IAV in a non-permeabilized staining. Scale bars represent 10 µm. Data are expressed as mean ± SEM of n=6–7 lungs/group. All data were pooled from three independent experiments and described biological replicates. *p<0.05, **p<0.01. PLA, Proximity Ligation Assay. CTCF, Corrected Total Cell Fluorescence.

The online version of this article includes the following source data for figure 4:

**Source data 1.** PDF file containing original membrane for *Figure 4A*, indicating the relevant bands and treatments.

**Source data 2.** Original membrane corresponding to *Figure 4A*.

demonstrated that Plscr1 successfully pulled down Ifn-$\lambda$r1 in both uninfected and IAV-infected WT mouse lungs (*Figure 4A*). Since the co-IP assay may detect indirect interactions through intermediate partners, we used proximity ligation assay (PLA) as a more precise method to identify direct interactions. Consistently, using PLA (*Figure 4B*), we detected direct interactions between Ifn-$\lambda$r1 and Plscr1 on airway and alveolar epithelial cells in WT mouse lungs. Noteworthily, significantly more and stronger PLA signals per area were detected in IAV-infected lungs compared to uninfected lungs. In agreement with the PLA, in unpermeabilized Calu-3 cells, Plscr1, and Ifn-$\lambda$r1 colocalized on the cell membrane after IAV infection (*Figure 4C*). In contrast, neither colocalization nor clear expression of Plscr1 or Ifn-$\lambda$r1 was evident without infection, implying an infection-specific interaction in human airway epithelial cells (*Figure 4C*). Hence, Plscr1 may regulate Ifn-$\lambda$r1 expression and IFN-$\lambda$ signaling in airway epithelial cells through both gene transcription and protein interaction in IAV infection.

## Both cell surface and nuclear PLSCR1 regulates IFN-$\lambda$ signaling and limits IAV infection independent of its enzymatic activity

PLSCR1 mutants with specific PLSCR1 cellular distribution were employed to determine the relative contributions of cell surface and nuclear PLSCR1 in regulating IFN-$\lambda$ signaling and IAV infection. PLSCR1 contains a 5-cysteine palmitoylation motif ($C^{184}CCPCC^{189}$), where substitution of these cysteines with alanine (labeled 5CA) completely abolishes its membrane localization, leading to exclusive localization in the cytosol and nucleus (*Wiedmer et al., 2003*). On the other hand, a single amino acid mutation of histidine[262] to tyrosine (labeled H262Y) in the non-classical nuclear localization signal of PLSCR1 completely abates its nuclear localization, leaving PLSCR1 exclusively in the cytosol and on the cell membrane (*Chen et al., 2005*). In addition, given that PS externalization plays a role in cell death regulation (*Bassé et al., 1996*), we asked whether the enzymatic activity of PLSCR1 interferes with IFN-$\lambda$ signaling. Phenylalanine[281] in the β-barrel hydrophobic loop region of PLSCR1 has been shown to be important for $Ca^{2+}$-dependent PS exposure (*Chen et al., 2005*), and a substitution with alanine (labeled F281A) renders PLSCR1 enzymatically inactive (*Xu et al., 2023*).

*PLSCR1(WT)*, *PLSCR1(5CA)*, *PLSCR1(H262Y)* and *PLSCR1(F281A)* plasmids built on PLV-EF1a-IRES-Hygro backbone were amplified in DH5α competent cells, co-transfected with HIV lentivector into 293T cells, and transduced into *Plscr1⁻ᐟ⁻* A549 cells. After a 10 day hygromycin selection, transduction efficiency and subcellular locations of PLSCR1 were analyzed using flow cytometry (*Figure 5—figure supplement 1A*). *PLSCR1* (WT) exhibited higher PLSCR1 expression than *Plscr1⁺ᐟ⁺* cells, indicating overexpression resulting from transduction (*Figure 5—figure supplement 1B*). Consistent with previous findings, *PLSCR1(5CA)* exhibited low surface expression, and *PLSCR1(H262Y)* had low nuclear expression of PLSCR1. *PLSCR1(F281A)* showed a similar distribution compared to *PLSCR1*(WT), confirming that the loss of enzymatic activity did not restrict the subcellular localization. Transduction efficiency ranged between 37 to 70% (*Figure 5—figure supplement 1C*).

In *PLSCR1⁺ᐟ⁺* cells or cells transduced with *PLSCR1*(WT), *PLSCR1(5CA)* or *PLSCR1(F281A)*, IFN-$\lambda$R1 expression was significantly induced by IAV infection. We found that PLSCR1 nuclear import was required for IFN-$\lambda$R1 expression, as *IFNLR1* transcription and protein translation was not induced in cells transduced with *PLSCR1(H262Y)* or in *PLSCR1⁻ᐟ⁻* cells (*Figure 5A and B*). In contrast, minimal protein interactions between PLSCR1 and IFN-$\lambda$R1 were detected by PLA in cells transduced with *PLSCR1(5CA) or in PLSCR1⁻ᐟ⁻ cells,* while *PLSCR1⁺ᐟ⁺* cells and cells transduced with *PLSCR1*(WT), *PLSCR1(H262Y)*, and *PLSCR1(F281A)* had strong protein binding of PLSCR1-IFN-$\lambda$R1, demonstrating that PLSCR1 interacts with IFN-$\lambda$R1 on cell membrane (*Figure 5C*). Furthermore, the lipid scramblase activity of PLSCR1 is uncoupled from its regulation of IFN-$\lambda$ signaling.

When compared to *PLSCR1⁺ᐟ⁺* cells, *PLSCR1*(WT), *PLSCR1(5CA)*, and *PLSCR1(F281A)* provided similar protection against IAV infection. However, *PLSCR1 (H262Y)* showed only partial viral clearance, as demonstrated by viral copy numbers measured by M gene segment qRT-PCR quantification (*Figure 5D*) and infectious viral titers from plaque assays (*Figure 5E*). This is further validated with a cell coverage assay when a high dose IAV was used for infection. Since the proliferation between differently transduced cell lines was similar and all cells were grown to full confluency before infection, the cell coverage represented resistance to infection and survival of cells. While *PLSCR1(5CA)* and *PLSCR1(F281A)* had a high surface coverage comparable to *PLSCR1*(WT), *PLSCR1(H262Y)* lost about half of the coverage (*Figure 5F*). Therefore, nuclear PLSCR1 is both essential and sufficient for IAV control through *IFNLR1* transcription regulation. Nevertheless, *PLSCR1(H262Y)* still rescued a

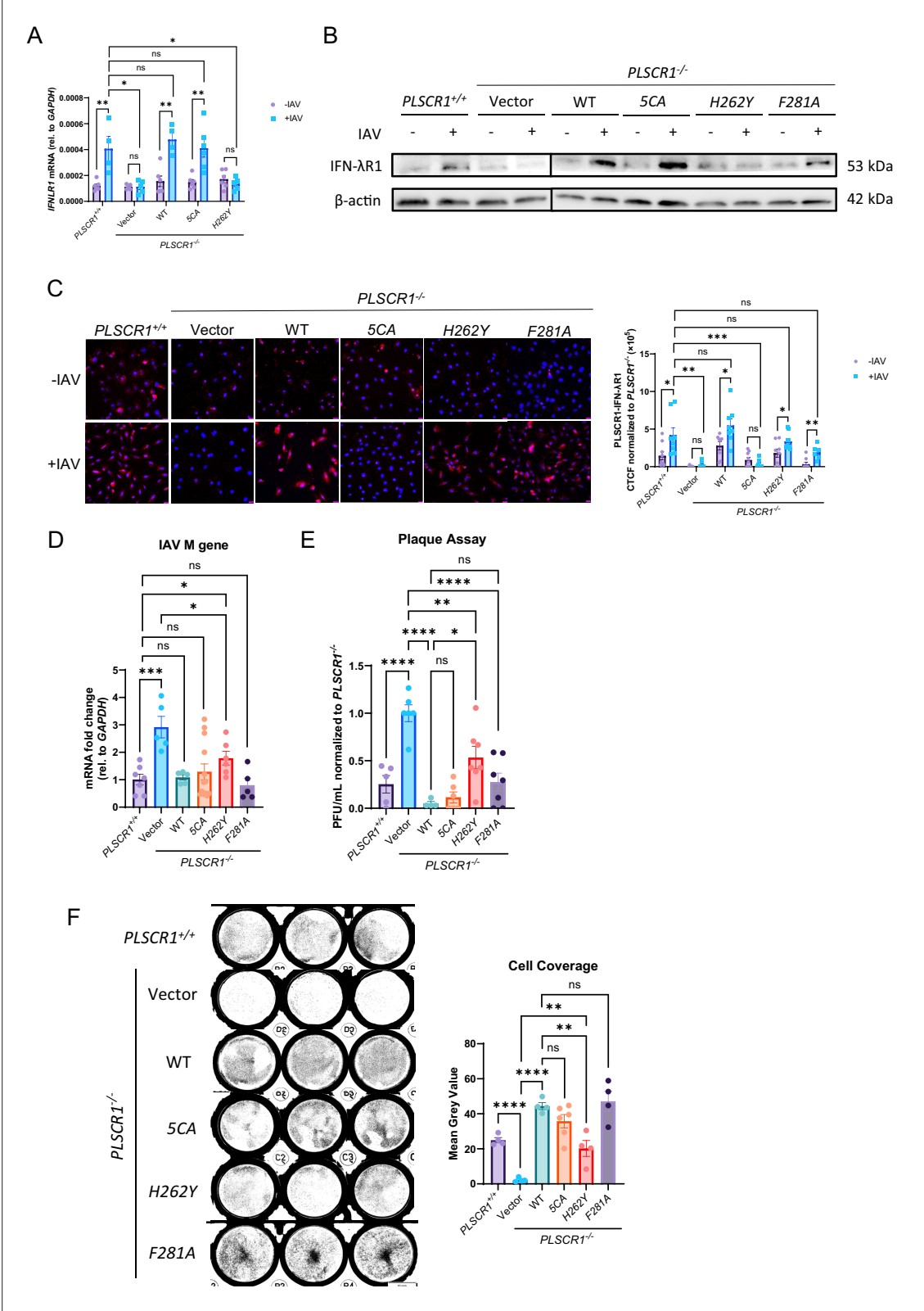

**Figure 5.** Requirement of both nuclear and surface PLSCR1 but not the enzymatic activity in IFN- λ R1-mediated anti-influenza activities. *PLSCR1⁻/⁻* A549 cells were transduced with mutated *PLSCR1* plasmids using lentivirus and infected with influenza A virus (IA)V (PR8) for 24 hr at 1 MOI (**A–E**) or 10 MOI (**F**). (**A**) *IFNLR1* RNA by qRT-PCR. (**B**) IFN- λ R1 proteins by western blotting. (**C**) Proximity ligation assay of IFN- λ R1 and PLSCR1. Scale bars represent 20 μm. Quantifications were performed using ImageJ. (**D**) Viral RNA load was assessed by quantifying M gene by qRT-PCR. (**E**) Infectious viral titer was

*Figure 5 continued on next page*

*Figure 5 continued*

assessed by plaque assays. (**F**) Cells were stained with crystal violet. Cell viability was quantified using ImageJ. Data are expressed as mean ± SEM of n=4–13 wells/group. All data were pooled from three independent experiments and described biological replicates. ns, not significant, *$p<0.05$, **$p<0.01$, ***$p<0.001$, ****$p<0.0001$, *****$p<0.00001$. CTCF, Corrected Total Cell Fluorescence.

The online version of this article includes the following source data and figure supplement(s) for figure 5:

**Source data 1.** PDF file containing original membrane for *Figure 5B*, indicating the relevant bands and treatments.

**Source data 2.** Original membrane corresponding to *Figure 5B*.

**Figure supplement 1.** PLSCR1 transduction efficiency and distribution.

significant number of cells when reintroduced into *PLSCR1$^{-/-}$* cells, indicating a partial protective role of PLSCR1 on epithelial cell surface through IFN-$\lambda$R1 protein regulation. Moreover, the anti-IAV function of PLSCR1 is independent of its lipid scramblase activity. Importantly, our findings that the scramblase activity is not required for protection against IAV infection is consistent with earlier observations with SARS-CoV-2 (*Xu et al., 2023*), suggesting that the scramblase activity of PLSCR1 may generally be dispensable for protection against major human viruses.

## The anti-influenza activity of Plscr1 is highly dependent on Ifn-$\lambda$r1

To establish a causal link between the impaired type 3 IFN pathway and the increased susceptibility to IAV observed in *Plscr1$^{-/-}$* mice and *PLSCR1$^{-/-}$* cells, *Ifnlr1$^{-/-}$* mice were crossed with *Plscr1$^{-/-}$* mice to generate double-knockout *Plscr1$^{-/-}$;Ifnlr1$^{-/-}$* mice. Immunofluorescence confirmed the absence of Ifn-$\lambda$r1 expression in *Ifnlr1$^{-/-}$* mice (*Figure 6A*). Following sublethal IAV infection, *Ifnlr1$^{-/-}$* mice exhibited significantly greater body weight loss at 3 dpi than *Plscr1$^{-/-}$* mice, which retain basal Ifn-$\lambda$r1 expression (*Figure 6B*). Moreover, they also showed significantly higher total BAL cell counts (*Figure 6C*), increased neutrophil percentages (*Figure 6D*), and higher infectious viral titers as measured by plaque assays (*Figure 6E*). These findings indicate that complete loss of Ifn-$\lambda$r1 results in greater susceptibility to IAV than the loss of Plscr1-mediated Ifn-$\lambda$r1 upregulation, underscoring the essential role of the type 3 IFN pathway in anti-viral defense.

Importantly, *Plscr1$^{-/-}$;Ifnlr1$^{-/-}$* mice showed no further increase in weight loss (*Figure 6B*), total BAL cell counts (*Figure 6C*), neutrophil percentages (*Figure 6D*), and IAV titers (*Figure 6E*) compared to *Ifnlr1$^{-/-}$* mouse lungs, indicating that the antiviral activity of Plscr1 is largely dependent on Ifn-$\lambda$r1.

## PLSCR1 expression is upregulated in the ciliated airway epithelial compartment of mice following flu infection

To understand the specific cell types that have increased PLSCR1 expression following flu infection in vivo, we explored a comprehensive scRNA sequencing dataset generated from uninfected or IAV-infected mouse lungs at 0, 1, 3, 6, and 21 dpi (*Boyd et al., 2020*). Distinct lung cell populations were identified using the Seurat algorithm. We assigned a total of 38 clusters based on their transcriptomic signatures, including 13 epithelial populations, 5 endothelial populations, 9 mesenchymal populations, and 11 immune populations (*Supplementary file 2*). Two-dimensional UMAP (*Figure 7A*) and bar charts of the proportion (*Figure 7—figure supplement 1A*) and cell count (*Figure 7—figure supplement 1B*) of each cluster demonstrated that these main lung cell populations (epithelial, endothelial, mesenchymal, and immune) were dynamic over the course of infection. Following infection, many populations emerged, particularly within the immune cell clusters. At the same time, some clusters were initially depleted and later restored, such as microvascular endothelial cells (cluster 2). Other populations, such as interferon-responsive fibroblasts (cluster 20), showed a dramatic yet transient expansion during acute infection and disappeared after infection resolved.

Within the epithelial populations, *Plscr1* was mainly expressed by ciliated epithelial cells, mesothelial epithelial cells, Epcam +Pecam1+ cells, club cells, and AT1 cells (in decreasing order of aggregated expression, *Figure 7B*). Conversely, AT2 cells, epithelial-mesenchymal transitional cells, and Krt8 + cells exhibited very low levels of *Plscr1* expression. Among the various epithelial populations, ciliated epithelial cells exhibited both the highest aggregated expression of *Plscr1* (*Figure 7B*) and the most significant upregulation ($p<2.22e-16$ and $p=6.7e-05$) at 3 dpi in early IAV infection (*Figure 7C*, *Figure 7—figure supplement 2A–K*). In contrast, AT1 cells were the only other epithelial cluster to show *Plscr1* upregulation at 3 dpi, but to a much less extent ($p=0.033$, *Figure 7—figure supplement*

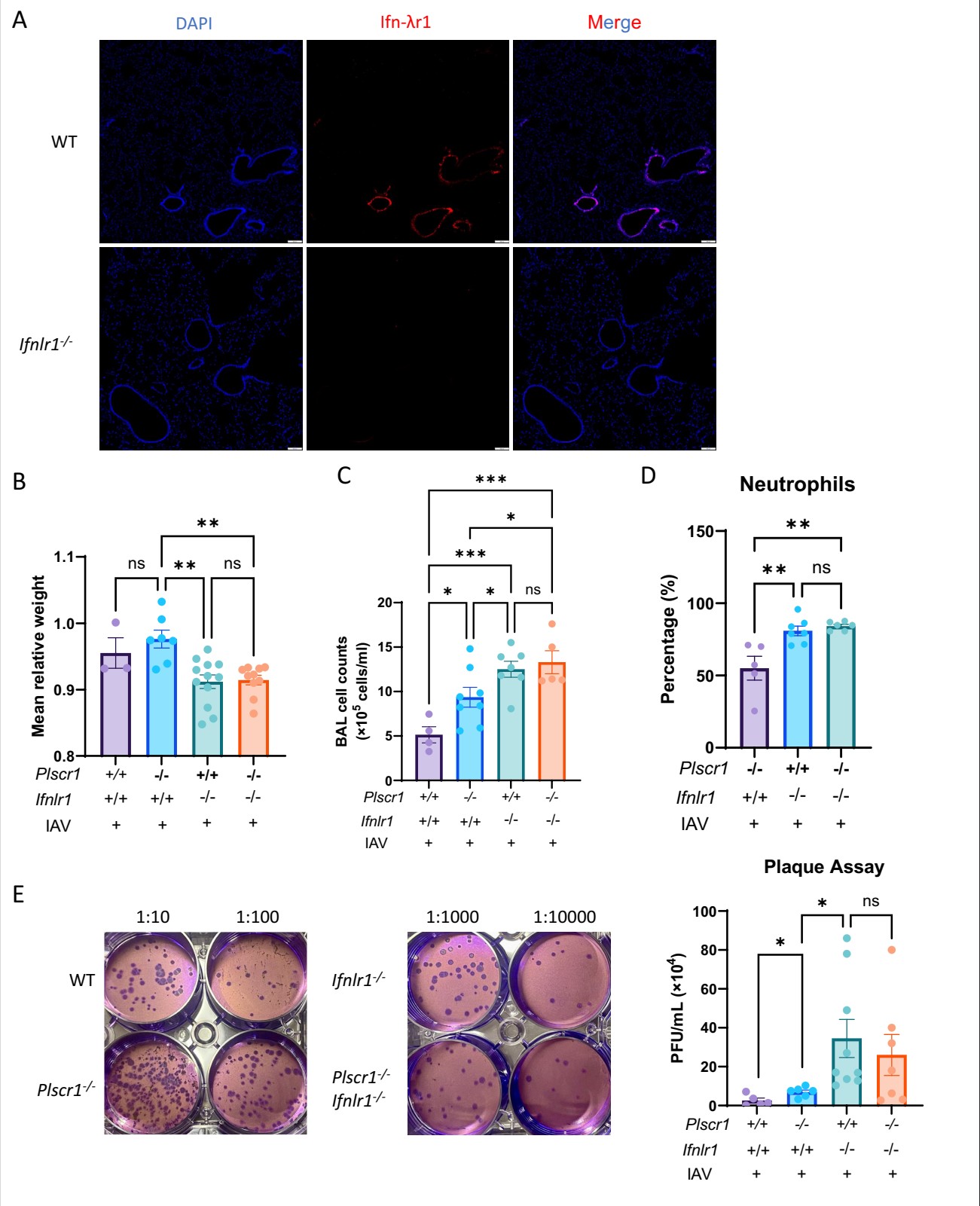

**Figure 6.** The relative contribution of the Type 3 IFN pathway to Plscr1-mediated antiviral immunity. WIld-type (WT), *Plscr1⁻/⁻*, *Ifnlr1⁻/⁻* and *Plscr1⁻/⁻;Ifnlr1⁻/⁻* mice were exposed to sublethal (300 pfu) influenza A virus (IAV) (WSN) infection and sacrificed at 3 dpi. (**A**) Representative immunofluorescent staining for DAPI and Ifn- λ r1 in lungs. Scale bars represent 100 μm. (**B**) Mean relative weight of mice. (**C**) Total Bronchoalveolar lavage (BAL) leukocyte numbers. (**D**) Neutrophil percentages in BAL. (**E**) Infectious viral titer in the lungs was assessed by plaque assays. Data are expressed as mean ± SEM of n=3–

*Figure 6 continued on next page*

*Figure 6 continued*

12 mice/group. All data were pooled from three independent experiments and described biological replicates. ns, not significant, *p<0.05, **p<0.01, ***p<0.001.

---

*2J*). Moreover, the increased expression of Plscr1 persisted for at least 21 days in ciliated epithelial cells, suggesting that Plscr1 may primarily exert its anti-flu activities in these cells (*Figure 7C*).

For the predominant ciliated epithelial cell cluster, we performed Gene Ontology (GO) analysis of the differentially expressed genes to compare potentially distinct functions before and after infection (*Figure 7D*). We found that the most upregulated pathways at 1 dpi were related to protein folding and localization, indicating a rapid cellular response to infection by altering their protein profile. At 21 dpi, ciliated epithelial cells were characterized by adaptive immune regulation, such as antigen presentation and memory cell generation. At 3 and 6 dpi, ciliated epithelial cells exhibited similar innate immune and inflammatory signatures dominated by interferon signaling. As expected, *Plscr1* participated in all of the top 10 enriched pathways, implying that it could facilitate robust antiviral responses in ciliated epithelial cells as an ISG during acute infection.

Gene expression levels of the most differentially expressed genes at individual time points in ciliated epithelial cells were shown in a heatmap (*Figure 7E*). Signature genes for cells at homeostasis included *Jund* (JunD proto-oncogene), *mt-Nd3* (mitochondrial NADH dehydrogenase 3), and *Cdkn1c* (cyclin-dependent kinase inhibitor 1 C), which are regulators of cell metabolism and proliferation (*Meixner et al., 2010*; *Kraja et al., 2019*; *Borges et al., 2015*). Signature genes enriched at 1 dpi were *Hspa5* (heat shock protein family A member 5), *Hsph1* (heat shock protein family H member 1), and *Clu* (clusterin), which play critical roles in protein folding and assembly (*Hendershot et al., 1994*; *Mattoo et al., 2013*; *Humphreys et al., 1999*). Consistent with the GO analysis, ciliated cells at 3 and 6 dpi had very similar signature genes associated with interferon pathways, namely a large component of ISGs, such as *Isg15* (interferon-stimulated gene 15), *Oasl2* (2'–5'-oligoadenylate synthetase like 2) and *Irf7* (interferon regulatory factor 7). Importantly, Plscr1 is required to potentiate the expressions of many of these ISGs (*Dong et al., 2004*), highlighting its potentially multifunction in anti-flu responses of ciliated epithelial cells. Signature genes for 21 dpi largely overlapped with those for 10 dpi, containing many histocompatibility 2, class II antigen proteins, which are mediators of MHC class II antigen processing and presentation.

In contrast to epithelial clusters, immune populations had low levels of *PLSCR1*, with relatively high expressions observed in alveolar macrophages, NK cells, and regulatory T cells (*Figure 7F*). However, we did not observe any significant difference in PLSCR1 levels before and post-infection in macrophages and neutrophils (*Figure 7G and H*), the two immune populations that express IFN-$\lambda$ R1 besides dendritic cells (*Andreakos et al., 2019*). Hence, the anti-influenza activities we described are unlikely to be carried out by immune cells.

## Overexpression of Plscr1 in ciliated epithelial cell, but not myeloid cell, is sufficient to provide anti-flu protection through IFN-$\lambda$ signaling

To further validate the relevant contributions of ciliated epithelial cells versus myeloid populations, including macrophages and neutrophils, we generated *Rosa26* locus-targeted *Plscr1* conditional knock-in transgenic mice (*Rosa26-LoxP-STOP-LoxP-Plscr1*; labeled *Plscr1$^{floxStop}$*). These mice were bred with *Foxj1-Cre* or *Lyz2-Cre* mice to overexpress Plscr1 in ciliated epithelial cells or myeloid cells, respectively (*Figure 8A*). Overexpression was confirmed by qRT-PCR with whole lung lysate and immunofluorescence staining at baseline (*Figure 8B and C*, *Figure 8—figure supplement 1A*).

At 3 dpi with sublethal IAV infection, *Plscr1$^{floxStop}$;Foxj1-Cre$^+$* mice lost less body weight (*Figure 8D*). Moreover, they had lower viral copy numbers as measured by IAV M gene segment qRT-PCR (*Figure 8E*), and lower infectious viral titer measured by plaque assays (*Figure 8F*) compared to *Plscr1$^{floxStop}$* mice, indicating overexpression of Plscr1 in ciliated epithelial cells provides protection against flu infection. Concurrently, they had significantly lower total BAL cell counts (*Figure 8G*), lower neutrophil percentages (*Figure 8H*) and lower type 1 and 3 *IFN* expressions (*Figure 8J*). We then examined Ifn-$\lambda$ r1, and found that its transcription and protein expression were further increased in *Plscr1$^{floxStop}$;Foxj1-Cre$^+$* mice during acute infection (*Figure 8I*). Therefore, ciliated epithelial cell-specific overexpression of Plscr1 is sufficient to provide anti-influenza protection through IFN-$\lambda$ signaling. In contrast, using similar approaches, we found that overexpression of Plscr1 in myeloid cells did not protect mice from

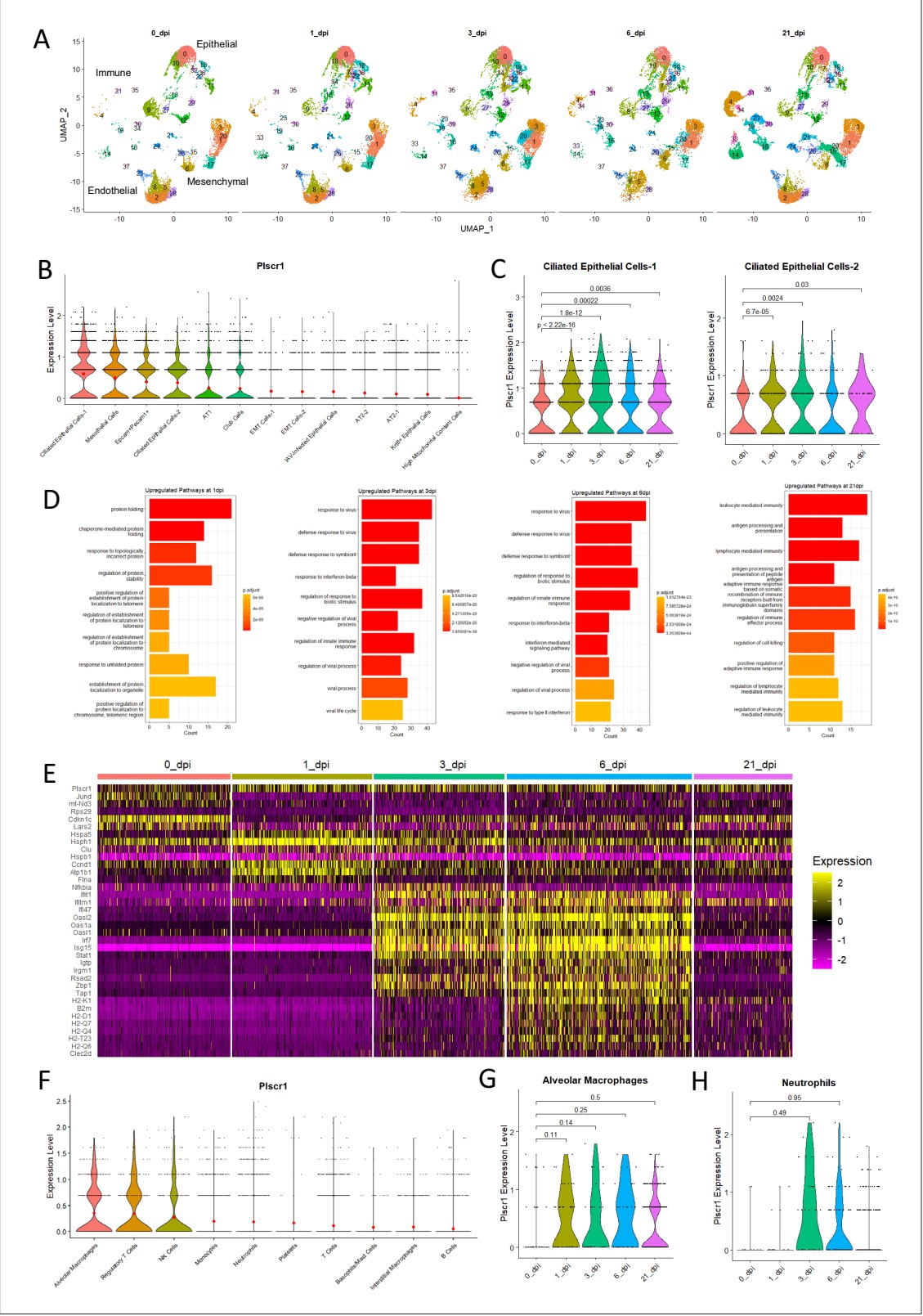

**Figure 7.** Cell-specific roles of *Plscr1* in influenza virus infection in mice. Wild-type (WT) mice were exposed to 2500 EID50 IAV (PR8) infection. Lungs were used for single-cell RNA sequencing analysis at 0, 1, 3, 6, and 21 dpi. (**A**) Two-dimensional UMAP representation of individual cells obtained from different timepoints. (**B**) Violin plot of aggregated *Plscr1* expressions in all epithelial cell clusters. Red dots represent mean expression levels. (**C**) Violin plot of time-dependent *Plscr1* expressions in both ciliated epithelial cell clusters. (**D**) Gene Ontology (GO) analysis for upregulated pathways

*Figure 7 continued on next page*

*Figure 7 continued*

in Ciliated Epithelial Cells-1. (**E**) Heatmap of the most differentially expressed genes of Ciliated Epithelial Cells-1 at different timepoints. (**F**) Violin plot of aggregated *Plscr1* expressions in all immune cell clusters. Red dots represent mean expression levels. (**G**) Violin plot of time-dependent *Plscr1* expressions in alveolar macrophage cluster. (**H**) Violin plot of time-dependent *Plscr1* expressions in neutrophil cluster.

The online version of this article includes the following figure supplement(s) for figure 7:

**Figure supplement 1.** Proportion and cell count of each cluster in single-cell RNA sequencing.

**Figure supplement 2.** Time-dependent *Plscr1* expressions in all epithelial cell clusters other than ciliated epithelial cells.

IAV infection: *Plscr1^floxStop^;Lyz2-Cre^+^* mice exhibited comparable weight loss, total BAL cell counts, viral copy numbers, histopathology, and expressions of interferons, *Ifnlr1,* and *Plscr1* at both early and late time points examined (***Figure 8—figure supplement 1B–I***).

Taken together, consistent with the scRNA sequencing, transgenic mouse models with cell-specific overexpression of Plscr1 demonstrated that Plscr1 offers antiviral protection in ciliated airway epithelial cells through IFN-$\lambda$ signaling (***Figure 9***).

## Discussion

We are the first group to demonstrate the roles of Plscr1 in a mouse-adapted human IAV-infected mouse model, to implicate its IFN-$\lambda$ signaling-related mechanisms, and to elucidate the cell types that are responsible for Plscr1-mediated anti-influenza activities. We established *Plscr1^-/-^* mice and found them more susceptible to IAV (WSN) compared to WT mice, as evidenced by greater weight loss in both sublethal and lethal infection and poorer survival in a lethal infection. Further examination of infected lungs provided the first in vivo evidence demonstrating that Plscr1 suppressed human IAV replication. This observation aligns with a previous report indicating that PLSCR1 interacts with the IAV NP, thereby impairing its nuclear import in vitro (***Luo et al., 2018***). Notably, while differences in viral copy numbers were only observed at the early stages of infection, coinciding with a significant increase in *Plscr1* transcription, these changes had profound implications for host fitness. Therefore, as one of the earliest induced ISGs, Plscr1 constitutes the frontline defense against influenza infection.

While the only previously published *Plscr1^-/-^* mouse flu model focused on an H1N1 SIV infection (***Liu et al., 2022***), our data showed both similarities and discrepancies. First, while both studies observed that Plscr1 promoted survival during IAV infection, SIV-infected *Plscr1^-/-^* mice exhibited weight loss similar to WT mice. Furthermore, while both models attributed the lower survival rate in *Plscr1^-/-^* mice to increased viral replication, SIV-infected *Plscr1^-/-^* lungs exhibited higher viral titers across all examined time points, from 1 to 7 dpi. Intriguingly, contrary to our observations, *Plscr1* expression was markedly decreased in SIV infection. Given previous in vitro studies demonstrating PLSCR1 induction by IAV (WSN) (***Luo et al., 2018***) and type 1 IFNs (***Zhou et al., 2000***; ***Dong et al., 2004***; ***Lizak and Yarovinsky, 2012***), we propose that the contradictory trend observed by Liu et al. may be attributed to distinct properties of SIV, such as viral replication rate, both the cellular tropism and the tissue tropism (proximal or distal lung), or antigen variation which may affect direct interaction with PLSCR1, innate sensing of the infection, or recognition by the adaptive immune response.

The delicate balance between immunity and immunopathology plays a pivotal role in determining host fitness during viral infections. To interrogate immunopathology in the lungs, we accessed the BAL, histology, and interferon expressions. BAL from *Plscr1^-/-^* mice were highly enriched with inflammatory neutrophils and lymphocytes, which were likely attracted by robust IFNs and other chemokines. Consistently, *Plscr1^-/-^* mice exhibited more severe lung damage and a greater extent of affected areas. These findings indicate that Plscr1 not only enhances immunity but also mitigates immunopathology. Importantly, regardless of excessive production of antiviral IFNs in *Plscr1^-/-^* mice, they failed to effectively control the initial viral infection. This suggests that the absence of Plscr1 impairs the IFN signaling pathway, highlighting the crucial role of Plscr1 in facilitating effective antiviral responses.

Although type 1 and 3 IFNs may share similar downstream pathways, they rely on distinct receptors for signaling. Consistent with previous findings (***Sheppard et al., 2003***), Ifn-$\lambda$r1 was detected in respiratory epithelium, including ciliated epithelial cells, club cells, and AT2 cells during infection. Loss of Plscr1 impaired *Ifnlr1* transcription in IAV infection, with this transcriptional difference translating into protein expression. IFN-$\lambda$ is crucial for early viral control within the initial days of infection without igniting unnecessary inflammation and compromising host fitness (***Galani et al., 2017***).

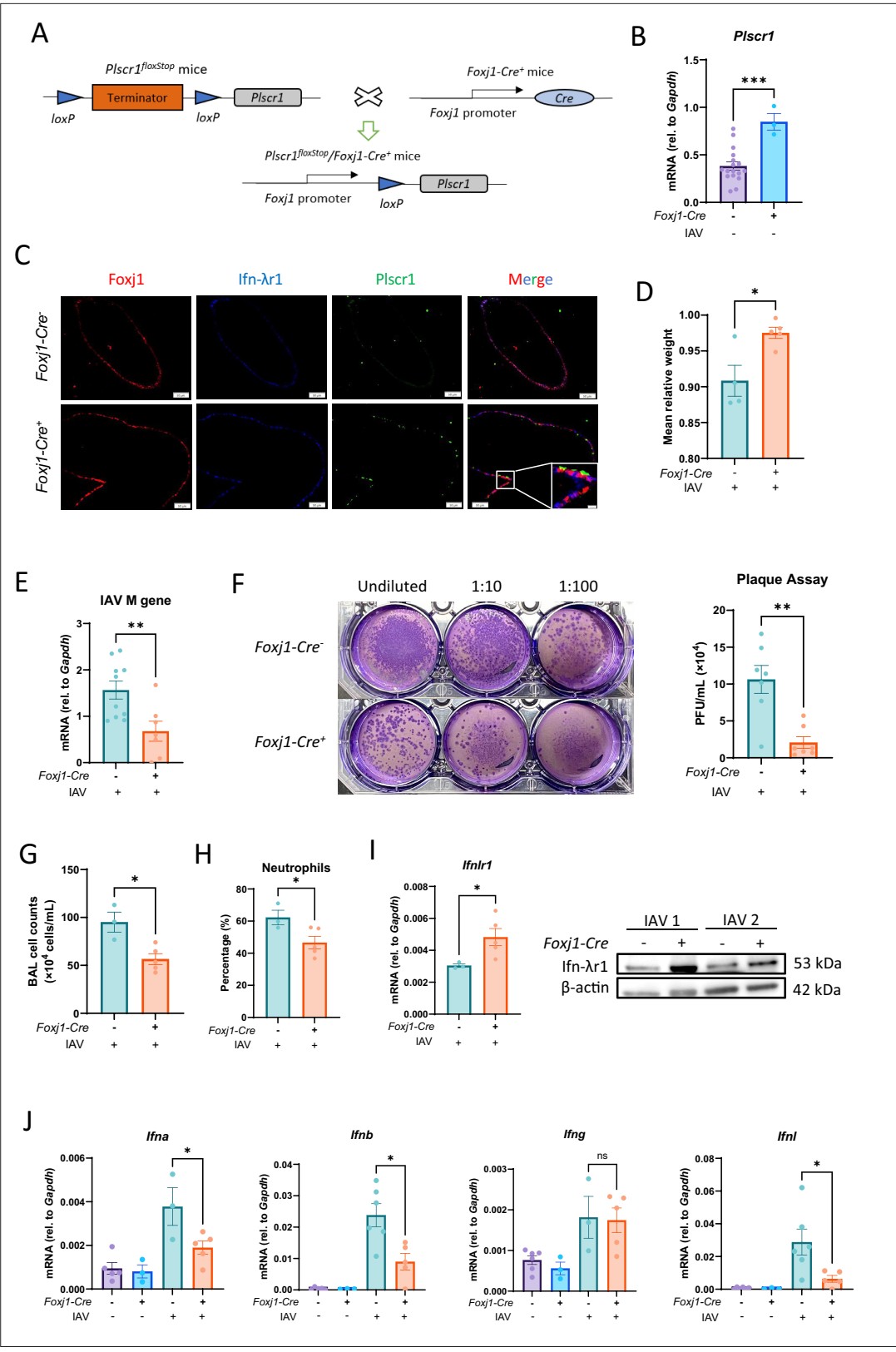

**Figure 8.** Reduced susceptibility of *Plscr1^floxStop^;Foxj1-Cre^+* mice to influenza virus infection. *Plscr1^floxStop^* and *Plscr1^floxStop^;Foxj1-Cre^+*mice were exposed to sublethal (300 pfu) influenza A virus (IAV) (WSN) infection and sacrificed at 3 dpi. (**A**) Schematic representation of the experimental design of ciliated epithelial cell conditional *Plscr1 KI* mice. (**B**) Validation of *Plscr1* overexpression in lungs of *Plscr1^floxStop^;Foxj1-Cre^+*mice by qRT-PCR.

*Figure 8 continued on next page*

*Figure 8 continued*

(**C**) Representative immunofluorescent staining for Plscr1, Ifn-λ r1, and Foxj1 in lungs. Scale bars represent 50 μm (main) and 10 μm (inlays).(**D**) Mean relative weight of mice. (**E**) Viral RNA load in the lungs was assessed by quantifying M gene by qRT-PCR. (**F**) Infectious viral titer in the lungs was assessed by plaque assays. (**G**) Total Bronchoalveolar lavage (BAL) leukocyte numbers. (**H**) Neutrophil percentages in BAL. (**I**) Whole lungs were analyzed for *Ifnlr1* RNA by qRT-PCR and Ifn-λ r1 protein by western blot. (**J**) Whole lungs were analyzed for *Ifna*, *Ifnb*, *Ifng*, and *Ifnl* RNA by qRT-PCR. (**K**) Model depicting proposed mechanism of PLSCR1-regulated IFN-λ signaling. Data are expressed as mean ± SEM of n=3–10 mice/group. All data were pooled from three independent experiments and described biological replicates. ns, not significant, \*p<0.05, \*\*p<0.01, \*\*\*p<0.001. dpi, days post-infection.

The online version of this article includes the following source data and figure supplement(s) for figure 8:

**Source data 1.** PDF file containing original membrane for *Figure 8I*, indicating the relevant bands and treatments.

**Source data 2.** Original membrane corresponding to *Figure 8I*.

**Figure supplement 1.** Unaffected susceptibility of *Plscr1^{floxStop}*;*Lyz2-Cre^+* mice to influenza virus infection.

With limited Ifn-λ r1 expression, *Plscr1^{-/-}* mice were unable to mount a robust type 3 IFN response to control early viral infection. Instead, they relied largely on type 1 interferons, which succeeded in eliminating IAV at later time points, but led to exaggerated immunopathology. Furthermore, our observations of enhanced neutrophilia, lung injury, and lethality in *Plscr1^{-/-}* mice align with findings reported in *Ifnlr1^{-/-}* mice in IAV infection (*Galani et al., 2017*). However, a discrepancy in *Ifnlr1* expression over the course of infection was observed between the RNA sequencing and the qRT-PCR data. While RNA-seq showed further upregulation of *Ifnlr1* at 7 dpi (*Figure 3A*), qRT-PCR indicated a rapid

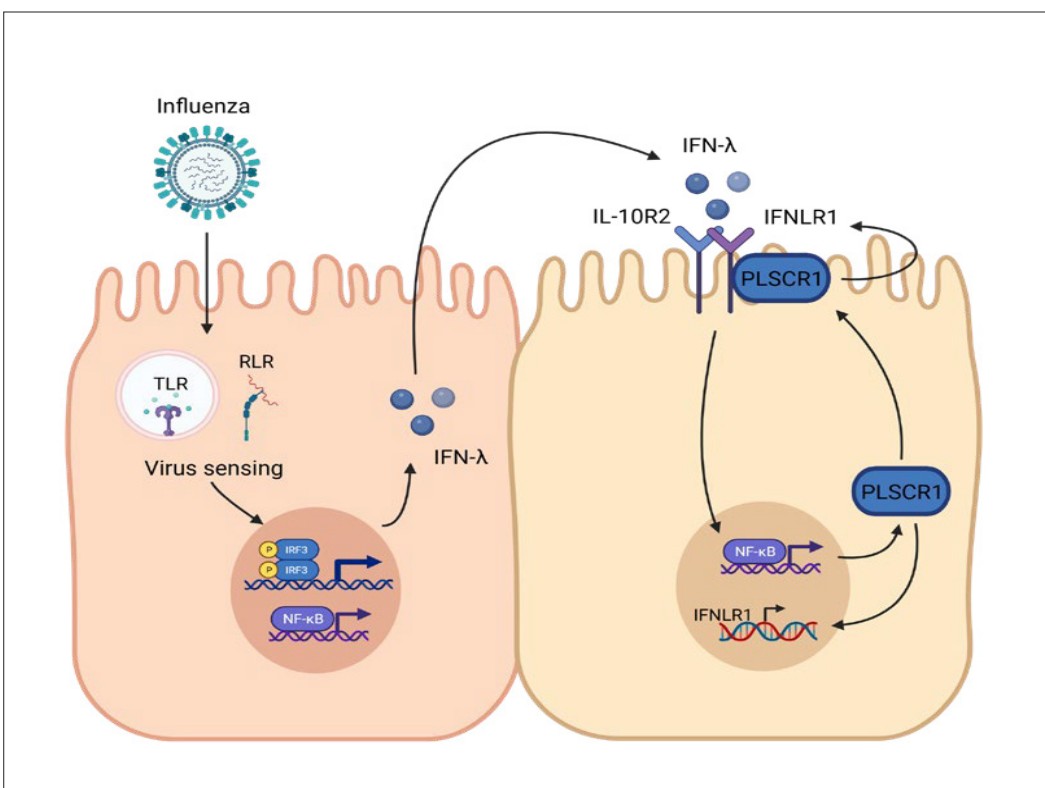

**Figure 9.** Model depicting proposed mechanism of PLSCR1-regulated IFN-λ signaling. Influenza infection is usually detected by intracellular pattern recognition receptors (PRRs) such as TLR 3 and 7, RIG-I, and MDA5. These PRRs activate the expression of *IFNL* in early infection stage through IRF-3 and NK-κB-controlled transcriptions. IFN-λ secreted by the infected cells interacts with IL-10R2 and IFN-λ R1 on neighboring cell surfaces, which results in activation of expression of various IFN-stimulated genes, including *PLSCR1*. In ciliated airway epithelial cells, PLSCR1 can further enhance the transcription of *IFNLR1* by directly binding to its promoter region as a transcriptional factor, or interact with IFN-λ R1 on the cell membrane.

downregulation at the same time point (*Figure 3B*). The reason for this time-dependent discrepancy remains unclear and warrants further investigation. In addition, this study did not definitively establish causality between reduced Ifn-$\lambda$ signaling and the observed in vivo phenotype. The increased morbidity and mortality in *Plscr1*[-/-] mice could also be attributed to elevated Tnf-α levels and associated lung damage. Given that proinflammatory cytokines and/or enhanced lung damage are known contributors to influenza morbidity and mortality, future work will be needed to disentangle the impacts of TNF-α, IL-1β, and other inflammatory cytokines from those of the IFN pathway to fully clarify the role of Plscr1 in antiviral defense.

PLSCR1 expression was increased in response to IFN-$\lambda$ in human airway epithelial cells, consistent with previous studies (*Xu et al., 2023*). While PLSCR1 typically localizes on the cell membrane and in the cytoplasm, it translocates into nucleus to bind the *IFNLR1* promoter upon IAV infection, thereby regulating *IFNLR1* transcription. The nuclear localization and functions of PLSCR1 have been extensively documented in previous studies (*Huang et al., 2020*; *Chen et al., 2013*; *Wyles et al., 2007*; *Huang et al., 2015*). Relevantly, IFN-α promotes the nuclear translocation of PLSCR1 in breast cancer cells (*Wiedmer et al., 2003*). Therefore, it is highly plausible that the nuclear trafficking of PLSCR1 in airway epithelial cells is similarly stimulated by IFN-α produced during IAV infection, but further evidence is demanded. Additionally, the precise binding site for PLSCR1 within the *IFNLR1* promoter and the binding motif on PLSCR1 remain unknown. Previous bioinformatics predictions revealed that the −430~−421 segment of *IFNLR1* promoter likely contains binding sites for a number of transcription factors (TFs) with important regulatory functions (*Ding et al., 2014*). Further mutagenesis studies, such as truncations or single-nucleotide mutations within these sequences, could be done to identify the specific motif for PLSCR1 binding. Finally, it is not clear whether PLSCR1 directly activates *IFNLR1* expression by acting as a TF, or as a co-factor enhancing other TF's transcriptional activity. Co-immunoprecipitations could be pursued in future to explore the binding potential between PLSCR1 and other known TFs for *IFNLR1*, such as NF-Y (*Ding et al., 2014*).

In addition to activities in the nucleus, PLSCR1 has been shown to interact with multiple proteins on the plasma or endosomal membrane (*Talukder et al., 2012*; *Guo et al., 2020*; *Sun et al., 2002*; *Li et al., 2006*; *Amir-Moazami et al., 2008*). Here, we reported a novel interaction between PLSCR1 and IFN-$\lambda$R1 on airway epithelial cell membrane in vivo and in vitro, confirmed with both coimmunoprecipitation and immunofluorescence. Since their interaction was significantly enhanced after IAV infection, we speculate that membrane-bound PLSCR1 is a positive regulator of IFN-$\lambda$R1 signaling. One plausible mechanism is that PLSCR1 facilitates the intracellular trafficking of IFN-$\lambda$R1, akin to its role in assisting the trafficking of other membrane receptors (*Talukder et al., 2012*; *Sun et al., 2002*; *Kametaka et al., 2003*).

The subcellular location of PLSCR1 is vital not only for interactions with host components, but also for direct viral control. We found that nuclear PLSCR1 is both necessary and sufficient for viral control in airway epithelial cells, whereas membrane PLSCR1 provides only partial protection against IAV infection. These findings are not surprising, as the previously reported anti-flu mechanism of PLSCR1 also relies on its nuclear localization signal to restrict the import of IAV NP (*Luo et al., 2018*). Furthermore, besides *IFNLR1*, PLSCR1 enhances the expression of a select subset of ISGs in IAV infection as well, a mechanism potentially mediated by nuclear PLSCR1 on ISG gene transcription (*Dong et al., 2004*). On the other hand, membrane PLSCR1 may modulate the JAK/STAT signaling pathway, thereby augmenting the optimal anti-viral activity of these ISGs (*Dong et al., 2004*). We found that membrane PLSCR1 interacts with IFN-$\lambda$R1 protein in IAV infection, suggesting that it could facilitate viral elimination to some extent.

PLSCR1 is most well-known for its scramblase activity that favors PS exposure, apoptosis, and phagocytosis (*Guo et al., 2020*; *Zhao et al., 1998*). Using an enzymatically inactive mutant of PLSCR1, we uncoupled its lipid scramblase activity from anti-influenza activity. There are several potential explanations for this finding. First, our epithelial cell culture lacked phagocytes, therefore, the impact of apoptosis followed by phagocytosis induced by PLSCR1 is minimal. Future studies using mice that harbor *Plscr1(F281A)* mutation would be needed to verify the role of lipid scramblase activity and epithelial cell apoptosis in the presence of phagocytes. Second, PLSCR1 exhibits only weak enzymatic activities compared to other members of lipid scramblase family, possibly due to its vastly different central β-barrel structure (*Xu et al., 2023*; *Tang et al., 2022*). PS externalization may be compensated by other more potent scramblases. Importantly, the lipid scramblase activity of PLSCR1 has

been shown to be dispensable for its anti-SARS-CoV-2 function in a similar manner (*Xu et al., 2023*), suggesting a general lack of significance for its enzymatic activity in viral infections.

Although PLSCR1 has several previously described anti-influenza functions, including interfering with viral nuclear import (*Luo et al., 2018*), regulating TLR9 signaling (*Talukder et al., 2012*), and potentiating the expression of other ISGs (*Dong et al., 2004*), our studies have clarified the relative contribution of the type 3 IFN pathway to Plscr1-mediated anti-influenza immunity using *Plscr1⁻/⁻;Ifnlr1⁻/⁻* mice. We observed that *Ifnlr1⁻/⁻* mice were more susceptible to IAV infection than *Plscr1⁻/⁻*, suggesting that the complete loss of Ifn-$\lambda$r1 results in worse protection than impaired Ifn-$\lambda$r1 upregulation alone. Moreover, the previously identified anti-IAV functions of Plscr1 do not appear sufficient to compensate for the loss of Ifn-$\lambda$r1 signaling in *Ifnlr1⁻/⁻* mice. The absence of further disease exacerbation or increased viral titers in *Plscr1⁻/⁻;Ifnlr1⁻/⁻* mice compared to *Ifnlr1⁻/⁻* mice indicates that the anti-influenza activity of Plscr1 is largely dependent on Ifn-$\lambda$r1.

While scRNA-seq analysis revealed that endothelial cells express Plscr1 most abundantly in the lung, they are not the major target of IAV infection, and IAV does not efficiently replicate in them (*Han et al., 2021*). Instead, airway epithelial cells are the frontline defense against respiratory pathogens, with ciliated epithelial cells being the only cell type that express α2,3-linked SA, the primary influenza virus receptor in the mouse airway (*Ibricevic et al., 2006*). Coincidently, our scRNA-seq results showed that ciliated epithelial cells not only had the highest aggregated expression of *Plscr1*, but also had the most significant increase in *Plscr1* expression in early IAV infection at 3 dpi. Experiments with *Plscr1^{floxStop};Foxj1-Cre⁺* mice further supported ciliated epithelial cell-dependent protection against IAV, with improved immunity and viral clearance, and dampened immunopathology as early as 3 dpi. These findings suggest that as a result of enhanced Ifn-$\lambda$r1 due to Plscr1 overexpression, type 3 interferons were able to exert their advantages being the earliest produced interferon, mounting both antiviral and anti-inflammatory responses in ciliated epithelial cells. To further establish the causal relationship between Plscr1 and Ifn-$\lambda$ signaling in airway ciliated epithelial cells, future experiments should focus on specifically overexpressing Plscr1 in ciliated epithelial cells on an *Ifnlr1⁻/⁻* background by breeding *Plscr1^{floxStop};Foxj1-Cre⁺;Ifnlr1⁻/⁻* mice. In addition, ciliated epithelial cells isolated from *Ifnlr1⁻/⁻* murine airways could be transduced with a *Plscr1* overexpression construct. We hypothesize that overexpression of Plscr1 in ciliated epithelial cells would not be able to rescue susceptibility in *Ifnlr1⁻/⁻* mice or cells, as the *Plscr1⁻/⁻;Ifnlr1⁻/⁻* mouse model suggests that Plscr1's Ifn-$\lambda$r1-independent anti-influenza mechanisms are likely minor compared to its role in upregulating Ifn-$\lambda$r1.

Taken together, our findings highlight the essential role of PLSCR1 in the regulation of IFN-$\lambda$R1 transcription in nucleus and expression on plasma membrane, both in vitro and in vivo. These mechanisms are crucial for inhibiting viral spread, reducing inflammation, and enhancing overall host fitness during IAV infection. Furthermore, we found that the enzymatic activity of PLSCR1 is dispensable for its anti-influenza function. Finally, ciliated airway epithelial cells are the primary cell type in the lung for mounting PLSCR1-mediated anti-influenza responses. The potential of PLSCR1 agonists that target ciliated airway epithelial cells as therapeutic treatments for influenza holds promise for future medical interventions. Moreover, our results have the potential to impact the classical yet evolving field of IFN signaling. Not only do these findings elucidate and expand our understanding of newly discovered IFN-$\lambda$ signaling, but they also shed light on the specific cell types and conditions under which IFN-$\lambda$ signaling is modulated. Given the significance of IFN-$\lambda$ signaling in various infectious diseases, these insights may pave the way for innovative therapeutic approaches targeting corresponding regulatory molecules in the treatment of other microbial infections in addition to influenza.

## Materials and methods

### Viruses

Purified A/WSN/1933 and A/PR/8/1934 (H1N1) virus (IAV) was kindly provided by Dr. Amanda Jamieson, Brown University. MDCK cells were used for the preparation of virus stocks and for virus titration by plaque assay.

## Cell culture

Human Calu-3 cells (ATCC) were kindly shared by Dr. Suchitra Kamle, Brown University. Cells were maintained in Eagle's Minimum Essential Medium (ATCC) with 10% fetal bovine serum (Gibco) and 1% penicillin-streptomycin (Sigma-Aldrich).

Madin-Darby canine kidney (MDCK) cells (ATCC) were kindly shared by Dr. Amanda Jamieson, Brown University. Cells were maintained in Dulbecco's Modified Eagle Medium (Genesee) with 10% fetal bovine serum and 1% penicillin-streptomycin.

Human A549 cells (ATCC) were kindly shared by Dr. John MacMicking, Yale University. Human 293T cells (ATCC) were kindly provided by Dr. Suchitra Kamle, Brown University. Both cells were maintained in Dulbecco's Modified Eagle Medium with 10% fetal bovine serum, 1% MEM non-essential amino acids (Gibco), and 1 mM sodium pyruvate (Gibco).

All cell lines used in this study were authenticated at ATCC by short tandem repeat (STR) profiling. In addition, all cell lines were routinely tested and confirmed negative for mycoplasma contamination using MycoAlert (Lonza) at regular intervals.

## Mice

Wild-type (WT) mice on a C57BL/6 J genetic background were purchased from the Jackson Laboratories. *Plscr1*$^{tm1a(EUCOMM)Hmgu}$ mice (*Plscr1*$^{-/-}$ mice on C57BL/6 background) were purchased from the International Mouse Phenotyping Consortium. Those mice have an FRT-flanked *lacZ*/neomycin sequence-tagged *LoxP* insertion upstream of *Plscr1* exon 6 and 7, and another *LoxP* insertion downstream of these critical exons. Subsequent *Cre* expression results in whole-body knockout mice that transcribed a shortened and nonfunctional transcript of *Plscr* (**Bult et al., 2019**). Genetic screening of *Plscr1*$^{-/-}$ mice was carried out by conventional PCR on genomic DNA from mouse tail biopsies using the following primers for *LacZ* reporter: Fw: 5'-GCGATCGTAATCACCCGAGT-3' and Rev: 5'-CCGCCAAGACTGTTACCCAT-3'. This set of primers generates a 307 bp fragment in *Plscr1*$^{-/-}$ mice.

*Rosa26* locus-targeted *Plscr1* conditional knock-in transgenic mice (*Rosa26-LoxP-STOP-LoxP-Plscr1 Tg; Plscr1*$^{floxStop}$ mice on C57BL/6 background) were generated at Brown Mouse Transgenic and Gene Targeting Facility. They were bred with *Lyz2-Cre* mice purchased from the Jackson Laboratories and *Foxj1-Cre* mice gifted by Drs. Yong Zhang and Michael Holtzman at Washington University School of Medicine in St. Louis to generate *Plscr1*$^{floxStop}$;*Lyz2-Cre*$^+$and *Plscr1*$^{floxStop}$;*Foxj1-Cre*$^+$mice, respectively. For the detection of *Cre* recombinase in *Lyz2-Cre* mice, the following primers were used: Common Fw: 5'-CTTGGGCTGCCAGAATTTCTC-3', Mutant Rev: 5'-CCCAGAAATGCCAGATTACG-3' and wild-type Rev: 5'-TTACAGTCGGCCAGGCTGAC-3'. A 700 bp fragment is amplified in homozygous mutants, while a 350 bp fragment is amplified in wild-type mice. Both fragments show up in heterozygotes. For the detection of *Cre* recombinase in *Foxj1-Cre* mice, the following primers were used: Mutant Fw: 5'-CGTATAGCCGAAATTGCCAGG-3' and Mutant Rev: 5'-CTGACCAGAGTCATCCTTAGC-3'. A 327 bp fragment is amplified in homozygous mutants.

*Ifnlr1*$^{tm1a(EUCOMM)WTsi}$; Deleter-Cre mice (*Ifnlr1*$^{-/-}$ mice on C57BL/6 background) were generated and gifted by Dr. Sanghyun Lee at Brown University (**Baldridge et al., 2017**; **Schwenk et al., 1995**). They were bred with *Plscr1*$^{-/-}$ mice to generate *Plscr1*$^{-/-}$;*Ifnlr1*$^{-/-}$ mice. For the detection of *Ifnlr1* in those mice, the following primers were used: Common Fw: 5'-AGGGAAGCCAAGGGGATGGC-3'; Rev 1: 5'-AGTGCCTGCTGAGGACCAGGA-3'; Rev 2: 5'-GGCTCTGGACCTACGCGCTG-3'. A 564 bp fragment is amplified in homozygous *Ifnlr1*$^{-/-}$ mutants, while a 231 bp fragment is amplified in wild-type mice.

All mice were housed and further bred in Brown University animal facilities. The mice with null mutation or overexpression of Plscr1 did not show any apparent abnormal phenotypes and developmental and signaling issues. This study was performed in strict accordance with the recommendations in the Guide for the Care and Use of Laboratory Animals of the National Institutes of Health. All of the animals were handled according to approved institutional animal care and use committee (IACUC) protocols (#24-09-0008) of Brown University. All terminal sacrifice was performed after euthanasia with urethane, and every effort was made to minimize suffering.

## Infection and treatment of mice

10–12 week-old mice were intranasally infected once with various doses of IAV in 30 µL of sterile PBS (Gibco), or treated with 2.5 µg/g of body weight of poly(I:C) (Invivogen) every day for 6 days under isoflurane-based anesthesia. Poly(I:C) was warmed up in a water bath at 37 °C before administration.

Thirty µL of sterile PBS was administered to control mice. Only males were used in infections. Littermates were randomly assigned to experimental and control groups.

## BAL harvest and differential cell counts

Mice were euthanized with intraperitoneal injection of 300 µL of urethane (0.18 g/ml, Sigma-Aldrich). Bronchoalveolar lavage (BAL) of the whole lung was performed with a total of 1 mL ice-cold PBS via an incision at the trachea. An aliquot was stained with trypan blue solution (Gibco) for viability and cell number determination using TC20 automated cell counter (Bio-Rad). Samples were centrifuged at 2500 rpm for 5 min at 4 °C. Cells were pelleted onto glass slides via cytospin centrifugation at 700 rpm for 7 min and stained with Hema 3 (Fisher Scientific). Neutrophils, macrophages, lymphocytes, and eosinophils were counted under a light microscope.

## RNA isolation and qPCR

Harvested right lungs were immediately snap-frozen in liquid nitrogen and stored at –80 °C afterwards. Frozen lungs were homogenized in TRIzol Reagent (Invitrogen). Cells were washed with ice-cold PBS once, directly lysed in the culture dish with TRIzol Reagent and collected using a cell scraper. RNA isolation was performed with RNeasy Mini Kit (Qiagen). 0.5 µg of the isolated RNA was used for cDNA synthesis with iScript cDNA Synthesis Kit (Bio-Rad). Real-time quantitative PCR was performed with iTaq Universal SYBR Green Supermix (Bio-Rad). All primers used were listed in *Supplementary file 1*. Relative amounts of mRNA expression were normalized to the level of mouse *Gapdh* or human *GAPDH*.

## Immunohistology and immunofluorescence

For immunohistology, immediately after euthanasia, left lungs were inflated through the incision at the trachea with 10% buffered formalin and soaked in formalin at RT. They were then transferred to 70% ethanol prior to paraffin embedding, sectioning, and staining with H&E performed at Brown University Molecular Pathology Core. Lungs were scanned with a VS200 slide scanner (Evident).

For immunofluorescence of mouse lungs, unstained paraffin-embedded lung sections were rehydrated in xylene (Fisher Scientific) and decreasing concentrations of ethanol (Pharmco). They were then steamed in antigen retrieval buffer (Abcam) for 30 min and blocked with 1% normal goat serum (Abcam) for 30 min at RT. Tissues were stained overnight at 4 °C with goat anti-mouse Ifn-$\lambda$r1 polyclonal antibody (1:300 dilution, Invitrogen, cat #PA1-21360, lot #WL3462592C), rabbit anti-mouse uteroglobin polyclonal antibody (1:300 dilution, Santa Cruz Biotechnology, cat #sc-365992, lot #F0220), rabbit anti-mouse Spc polyclonal antibody (1:100 dilution, Santa Cruz Biotechnology, cat #sc-13979, lot #C2007), mouse anti-mouse Foxj1 monoclonal antibody (1:300 dilution, Invitrogen, cat #14-9965-82, lot #2712325), and/or rabbit anti-mouse Plscr1 polyclonal antibody (1:300 dilution, Proteintech, cat #11582–1-AP, no lot/clone #). Tissues were washed 3 times with PBS and stained with chicken anti-goat Alexa 594 (Invitrogen, cat #A21468, lot #2318436), goat anti-rabbit Alexa 488 (Invitrogen, cat #A11008, lot #2051237) and/or donkey anti-goat Alexa 405 (Invitrogen, cat #A48259, lot #XI353690) for 1 hr at RT in the dark. For visualization of IAV, tissues were stained overnight at 4 °C with goat anti-H1N1-FITC (1:500 dilution, US Biological, cat #I7650-05E, lot #L23091555 C23082906). After washing 3 times with PBS, sections were mounted with Vectashield mounting medium with or without DAPI (Vector Laboratories). Lungs were visualized under an APEXVIEW APX100 microscope (Evident).

For immunofluorescence of cell cultures, cells were seeded in Millicell EZ slides (Millipore Sigma). After washing with ice-cold PBS, 4% paraformaldehyde (Boston BioProducts) was added on ice for 10 min for fixation and removed. If necessary, 0.2% Triton X-100 (Sigma-Aldrich) was added for 10 min to permeabilize the cells on ice, followed by three times of PBS wash. Cells were blocked with 5% bovine serum albumin (Fisher Scientific) for 1 hr at RT. They were stained overnight at 4 °C with mouse anti-human PLSCR1 monoclonal antibody (1:50 dilution, R&D Systems, cat #MAB8137, clone #875327) and rabbit anti-human IFN-$\lambda$R1 polyclonal antibody (1:100 dilution, Invitrogen, cat #PA5-98608, lot #ZB4228832A). Cells were washed 3 times with PBS and stained with chicken anti-mouse Alexa 594 (Invitrogen, cat #A21201, lot #2482957) and goat anti-rabbit Alexa 488 for 1 hr at RT in the dark. After washing 3 times with PBS, slides were mounted with Vectashield mounting medium with DAPI and visualized under a Nikon ECLIPSE Ti microscope.

When necessary, fluorescent images were analyzed using ImageJ (*Schneider et al., 2012*). Areas of interest were selected using freehand selection. Area, mean gray value, and integrated density were measured. For H1N1 staining, uninfected lungs were used as background readings. For Ifn-$\lambda$r1 staining, a region without fluorescence was used as a background reading. Corrected total cell fluorescence (CTCF) was calculated using the following formula: CTCF = Integrated Density – (Area of selection X Mean fluorescence of background readings).

## Proximity ligation assay (PLA)

Duolink PLA fluorescence (Millipore Sigma) was performed according to the manufacturer's protocol. In brief, unstained paraffin-embedded lung sections were rehydrated, retrieved, and blocked with Duolink Blocking Solution for 60 min at 37 °C. They were then incubated with goat anti-mouse Ifn-$\lambda$r1 polyclonal antibody and rabbit anti-mouse Plscr1 polyclonal antibody overnight at 4 °C. Next, slides were incubated with anti-goat PLUS and anti-rabbit MINUS probes for 1 hr at 37 °C. For cell culture, A549 cells were seeded in Millicell EZ slides, fixed, and permeabilized. After blocking, cells were incubated with rabbit anti-human IFN-$\lambda$R1 polyclonal antibody and mouse anti-human PLSCR1 monoclonal antibody overnight at 4 °C. Next, cells were incubated with anti-mouse PLUS and anti-rabbit MINUS probes for 1 hr at 37 °C. Diluted ligase and polymerase were applied to samples for 30 min at 37 °C in order. Finally, slides were mounted with Duolink In Situ Mounting Medium with DAPI and imaged with an APEXVIEW APX100 microscope.

## IAV infection and rhIFN-$\lambda$1 treatment in cell culture

Calu-3 cells at 80–90% confluency were washed twice with Opti-MEM (Gibco) and infected with IAV (WSN) at 0.1 multiplicity of infection (MOI) in Opti-MEM. One hour after viral attachment at 4 °C, IAV supernatant was removed. Cells were washed once and incubated in fresh Opti-MEM for 23 hr at 37 °C.

A549 cells at 80–90% confluency were washed once with PBS and infected with IAV (PR8) at 1, 5, or 10 MOI in PBS supplemented with BSA, $CaCl_2$, and $MgCl_2$. One hour after viral attainment at 37 °C, IAV supernatant was removed. Cells were incubated in Opti-MEM with 10 μg/mL trypsin for 23 hr at 37 °C.

Calu-3 or A549 cells were treated with 10 or 100 ng/mL of recombinant human IL-29/IFN-lambda 1 Protein (R&D Systems) for 6 or 24 hr at 37 °C. In control Calu-3 groups, cells were first incubated with 1 μg/mL of anti-human interferon lambda receptor 1 neutralizing antibody (PBL Assay Science, cat #21885–1, clone #MMHLR-1) for 1 hr, and then treated with IFN-$\lambda$.

## Co-immunoprecipitation (Co-IP)

Cryopreserved mouse lungs were homogenized in Pierce RIPA buffer (Thermo Scientific). Protein concentrations were determined using Pierce BCA Protein Assay Kit (Thermo Scientific) according to the manufacturer's protocol. Catch and Release v2.0 Reversible Immunoprecipitation System (Millipore) was used according to the manufacturer's protocol. In brief, 500 μg of lung lysate, 2 μg of rabbit anti-human Plscr1 polyclonal antibody or negative control human IgG and 10 μL of antibody capture affinity ligand were added to the spin columns with resin slurry. The mixture was incubated on a rotator at 4 °C overnight. The unbound flow-through was collected by centrifuge. Bound proteins were washed three times and eluted using denaturing elution buffer (5% SDS, 8 M Urea, and 100 mM Glycine).

## Western blot

Cryopreserved mouse lungs were homogenized in Pierce RIPA buffer. Cells in culture were washed with ice-cold PBS, lysed in Pierce RIPA buffer for 10 min on ice, and then scraped into RIPA buffer. Protein concentrations were determined using Pierce BCA Protein Assay Kit according to the manufacturer's protocol. Twenty μg of each sample was loaded into 4–20% Mini-PROTEAN TGX Gels (Bio-Rad) and run for 1 hr 20 min at 125 V in Tris/Glycine/SDS running buffer (Bio-Rad). Gels were transferred using the Trans-Blot Turbo Transfer Pack (Bio-Rad). Membranes were incubated with rabbit anti-mouse/human Plscr1 polyclonal antibody (1:1000 dilution), rabbit anti-mouse/human Ifn-$\lambda$r1 polyclonal antibody (1:1000 dilution, ABclonal, cat #A10082, lot #0094910201), and/or mouse anti-mouse/human β-actin monoclonal antibody (1:2500 dilution, Santa Cruz Biotechnology, cat #sc-47778, lot #F1323)

overnight at 4 °C. After washing with TBST (Boston BioProducts), membranes were incubated with HRP-linked anti-rabbit IgG (1:5000 dilution, Cell Signaling, cat #70745, lot #30) or HRP-linked anti-mouse IgG (1:5000 dilution, Cell Signaling, cat #70765, lot #36) for 1.5 hr at RT. Finally, membranes were treated with SuperSignal West Pico PLUS Chemiluminescent Substrate (Thermo Scientific) and imaged with ChemiDoc Imaging Systems (Bio-Rad).

## Chromatin-immunoprecipitation (ChIP)

SimpleCHIP Enzymatic Chromatin IP (Cell Signaling) was performed according to the manufacturer's protocol. In brief, IAV-infected Calu-3 cells were fixed with formaldehyde to cross-link proteins to DNA. Next, chromatin was digested with Micrococcal Nuclease into 150–900 bp DNA/protein fragments. Nuclear membrane was broken by sonication. A portion of chromatin preparation was purified for DNA prior to immunoprecipitation to confirm the digestion by electrophoresis and concentration by OD260. One μg rabbit anti-mouse Plscr1 polyclonal antibody, negative control normal rabbit IgG, or positive control Histone H3 rabbit monoclonal antibody was added to 5 μg chromatin preparation. The complex co-precipitated and was captured by Protein G magnetic beads. After 4 low salt washes and 2 high salt washes, chromatin was eluted and cross-links were reversed with proteinase K. DNA was purified using spin columns and analyzed by both standard PCR and qRT-PCR. For qRT-PCR, percent input method was used according to the following formula: Percent Input = 2% $\times$ $2^{(C[T] \, 2\%Input \, Sample \, - \, C[T] \, IP \, Sample)}$.

## Transformation of competent cells

All *PLSCR1* plasmids were provided by Dr. John MacMicking's Lab at Yale University. Plasmid DNA concentrations were determined using a spectrophotometer. Fifty μl of DH5α competent cells (Invitrogen) and 50 ng of each plasmid was mixed and incubated for 30 min on ice. Then, the plasmid and competent cell mixtures were heat shocked for 45 s at 42 °C, and cooled on ice for 2 min. Super Optimal broth with Catabolite repression (SOC) media (homemade with 2% tryptone, 0.5% yeast extract, 10 mM NaCl, 2.5 mM KCl, 10 mM MgCl2, 10 mM MgSO4, and 20 mM glucose) was added to bacteria. The cells were grown in a shaking incubator at 200 RPM for 45 min at 37 °C. The mixtures were then plated on LB agar ampicillin plates and incubated at 37 °C overnight.

Single colonies were selected for each plate, inoculated in Super Optimal Broth (SOB) media (homemade with 2% tryptone, 0.5% yeast extract, 10 mM NaCl, 2.5 mM KCl, 10 mM MgCl2, and 10 mM MgSO4) and grown to OD600 of 0.6 at 37 °C for 12–18 hr, with vigorous shaking of 200 RPM. Cultures were then centrifuged at 6000 G for 15 min at 4 °C. Cell pellets were used for plasmid isolation.

## DNA plasmid isolation

DNA plasmids were isolated using the QIAfilter Maxi Cartridges and QIAGEN Plasmid Maxi Kit, according to the manufacturer's protocol. In brief, *PLSCR1*-transduced competent cell pellets were resuspended in buffer P1 containing RNase A solution and lyse blue. After adding Buffer P2, mixtures were incubated at RT for up to 5 min. Buffer P3 was then mixed with the lysates, transferred into the barrel of the QIAfilter cartridges and incubated at RT for 10 min. Cell lysates were filtered through the equilibrated QIAGEN tip by gravity. DNA was eluted with Buffer QF and precipitated with isopropanol by centrifuging at 15,000 G for 10 min. Pellets were air-dried for 10 min and dissolved in di-water. A spectrophotometer was used to determine plasmid concentrations.

## Lentiviral packaging

293T cells were grown in DMEM until 90–95% confluency and then switched to Opti-MEM for 2 hr. Ten μg of plasmid of interest was mixed with lipofectamine 300 transfection reagent (Invitrogen), pPACKH1 HIV Lentivector Packaging Kit (System Biosciences) and Opti-MEM, and incubated for 20 min at RT. After removing half media from the cell culture dishes, DNA-lipid complex was added and cells were incubated at 37 °C with 5% CO2 for 6 hr. After 6 hr, culture supernatants were replaced with viral harvesting media (DMEM with glutamine, 10% FBS, 1% MEM non-essential amino acids, 1 mM sodium pyruvate, 2% BSA, and 1:50 Hank's solution) for overnight incubation. Viral supernatant was collected every day for the next 4 days by centrifuging cell culture supernatant at 1200 G for 5 min. Viral supernatant was mixed with lentiviral concentrator solution (40% polyethylene glycol 8000

and 1.2 M NaCl in PBS) at 4:1, and centrifuged at 1600 G for 60 min at 4 °C. The viral pellets were resuspended with Opti-MEM and stored at –80 °C.

## Lentiviral transduction

*PLSCR1*[-/-] A549 cells were seeded in a 96-well plate with DMEM. Lentiviral stocks were serially diluted in DMEM and polybrene (Millipore Sigma) mixture from 1:10 to 1:10,000, and added to cell culture. The 96-well plate was centrifuged at 2000 RPM for 30 min at RT and incubated at 37 °C with 5% $CO_2$ overnight. Media was replaced with fresh DMEM without polybrene the next day. After 48 hr post-transduction, DMEM with 200 µg/ml hygromycin was added for antibiotic selection for 10 days.

## Flow cytometry

Antibodies were titrated prior to confirming transduction efficiency by flow cytometry. One million cells were used for each sample. Samples were stained with LIVE/DEAD fixable violet dead cell stain for 405 nm excitation (1:1000, Invitrogen, lot #2581659) for 30 min. Surface and cytoplasm staining samples were fixed with 4% paraformaldehyde for 20 min at RT. Nuclear staining samples were fixed with true nuclear fix concentrate (BioLegend) for 45 min at RT. All samples were stained with mouse anti-human PLSCR1 monoclonal antibody (1:500) for 20 min, and chicken anti-mouse Alexa 594 (1:1000) for 20 min on ice. After fixation and each antibody incubation, surface staining samples were washed twice with FACS buffer (1% bovine serum albumin), cytoplasm staining samples were washed twice with Perm/Wash buffer (BD), and nuclear staining samples were washed twice with true nuclear perm buffer (BioLegend). Final samples were resuspended in FACS buffer. Flow cytometry analysis was performed with FACSAria IIIu (BD) by Brown University Flow Cytometry and Sorting Facility.

## Cell coverage assay

After IAV infection, A549 cells in 12-well plates were fixed with 4% paraformaldehyde in PBS for 1 hr at RT. Cells were then stained with 1% crystal violet solution (Sigma-Aldrich) for 5 min, washed with di-water three times and air-dried. Plates were scanned with an APEXVIEW APX100 microscope. Brightfield images were converted to 8-bit and the threshold was adjusted using a dark background in ImageJ (*Schneider et al., 2012*). Areas of interest were selected using oval selection. Cell coverages were quantified by measuring mean gray values of each well.

## Plaque assay

For plaque assay of IAV titer in mouse lungs, a small piece of frozen lung was weighed and homogenized in 10 mL decarbonated DMEM (MP Biomedicals) per 1 g of lung. The homogenizer was sterilized in between samples with washes in the following order: twice with 0.1% SDS (Bio-Rad), 1:100 diluted bleach in di-water, once with 0.1% SDS, 0.001% Coomassie blue in di-water, twice with 70% ethanol, and once with sterile PBS. Homogenates were centrifuged at 2000 G for 5 min at RT, and supernatant was collected.

For plaque assay of IAV titer in cell cultures, flu supernatant was collected, centrifuged at 2000 G for 10 min, and filtered through 0.45 µm filter.

MDCK cells were grown in 6-well plates to ~80% confluency. Virus was diluted to desired concentrations with PBS supplemented with BSA, $CaCl_2$, and $MgCl_2$. Media was aspirated from MDCK cell cultures and replaced with 100 µL of viral samples in each well. The plates were incubated for 1hr at 37°C and agitated every 15 min to prevent from drying. After 1hrour, flu was aspirated and the wells were filled with 2 mL plaque assay overlay (DMEM/F-12 (Gibco) supplemented with $NaHCO_3$, BSA, DEAE dextran, and penicillin-streptomycin, mixed with 2% bacteriological agar (Oxoid) at 1:1 ratio). The plates were left at RT in the hood for 5 min to let the agar solidify, and then placed upside down in the 37 °C incubator for 3 days until visible plaques were observed. Cells were fixed with 4% paraformaldehyde at RT for 1 hr. Agar overlay was removed with a metal spatula. Cells were stained with crystal violet solution gram (Harleco) for 5 min, and washed three times with di-water. Plates were air-dried. IAV titer was determined using the following formula: PFU/mL = # of plaques×1/dilution factor×10.

## Bulk RNA sequencing

Selected RNA isolated from mouse lungs in the same treatment group were pooled together into one sample. Pooled RNA concentrations were measured using a Nanodrop (Thermo Scientific). Quality control, library preparation, bulk RNA sequencing, and data analysis was performed by Azenta Genomics/GENEWIZ. Heatmaps were made with expressions of selected genes in Morpheus (https://software.broadinstitute.org/morpheus). Hierarchical clustering was performed with one minus Pearson correlation, complete linkage method, and clustering by row. ISGs were identified using the Interferome database, with selections of a fold change ≥2 in all cell types in lung of *Mus musculus* (*Rusinova et al., 2013*).

Sequencing data have been deposited in GEO under accession codes GSE307868. Datasets Generated: Phospholipid Scramblase 1 (PLSCR1) Regulates Interferon-Lambda Receptor 1 (IFN-$\lambda$ R1) and IFN-$\lambda$ Signaling in Influenza A Virus (IAV) Infection: Yang AX, Ramos-Rodriguez L, Sorkhdini P, Yang D, Norbrun C, Majid S, Lee S, Zhang Y, Holtzman MJ, Boyd DF, Zhou Y, 2025, https://www.ncbi. nlm.nih.gov/geo/query/acc.cgi?acc=GSE307868, GEO, GSE307868.

## Single-cell RNA sequencing analysis

Single-cell RNA sequencing dataset (Spatial transcriptomics of mouse lung: Mouse D Section 1 at 10 days post-influenza infection: Boyd, D.F., Allen, E.K., Randolph, A.G. et al., 2020, https://www. ncbi.nlm.nih.gov/sra/?term=SRX8008853, NCBI Short Read Archive under BioProject PRJNA612345) was generated from uninfected or IAV-infected mouse lungs at 0, 1, 3, 6, and 21 dpi as previously described (*Boyd et al., 2020*). Single-cell transcriptomic libraries were generated using the 5′ Gene Expression Kit (V2, 10 X Genomics) according to the manufacturer's instructions with the addition of primers to amplify HTOs during cDNA amplification (*Boyd et al., 2020*). Sequencing was performed on the Illumina NovaSeq to generate approximately 500 million reads per sample (*Boyd et al., 2020*). The 10 X gene-expression data were processed and normalized using CellRanger (v.3.0.2, 10 X Genomics) (*Boyd et al., 2020*). QC was performed first by excluding any gene that was not present in at least 0.1% of total called cells and then by excluding cells that exhibited extremes in the species-specific distributions of: the number of genes expressed (<100 or >5550), the number of mRNA molecules (>30,000), or the percentage of expression owed to mitochondrial genes (>7.5%) (*Boyd et al., 2020*). Scaling, principal component analysis, dimensionality reduction and cell clustering were performed using the Seurat algorithm as previously described (*Boyd et al., 2020*; *Hao et al., 2021*). Cell clusters were annotated using known markers from the literature (*Supplementary file 2*).

All following analysis were performed using R Statistical Software (v4.3.0) (*R Development Core Team, 2023*). Two-dimensional UMAPs were generated with DimPlot (Seurat) (*Hao et al., 2021*). Bar charts of proportion and cell count of each cluster were plotted with geom_col (ggplot2) (*Wickham, 2016*). Violin plots of *Plscr1* expression in different cell clusters were plotted with VlnPlot (Seurat) (*Hao et al., 2021*). Differentially expressed genes for ciliated epithelial cell clusters at different timepoints in comparison to 0 dpi were identified with FindMarkers (Seurat) (*Hao et al., 2021*). Those genes were used for Gene Ontology (GO) analysis with enrichGo (clusterProfiler) (*Wu et al., 2021*), and heatmaps with DoHeatmap (Seurat) (*Hao et al., 2021*).

## Quantitation and statistical analysis

Data was analyzed on GraphPad Prism software. Log-rank (Mantel-Cox) test was used to compare survival rates. Ordinary two-way ANOVA tests were used to compare weight losses. Statistical significance of differences was assessed using the parametric Student's two-tailed t-tests for all other normally distributed data. Differences were considered significant when $p<0.05$. Outlier tests with the ROUT method (Q=1%) were used to identify and remove any outliers.

## Acknowledgements

The authors thank A Ayala, A Jamieson, and C Lee for advice; J MacMicking for sharing *PLSCR1*[-/-] and *T16F*[-/-] A549 cells and all *PLSCR1* plasmids; Brown University Molecular Pathology Core for histopathology; and Brown University Flow Cytometry and Sorting Facility for flow cytometry.

This work was supported by grants R01 HL146498 (YZ), P20 GM103652 (YZ), U54 GM115677 (YZ), T32 HL134625 (PS), AHA 24TPA1277918 (YZ), ATS 23-24PHP11 (YZ), R01 AI183155 (SL), and R35 HL145242 (MJH).

During the preparation of this work, the author(s) used ChatGPT developed by OpenAI in order to check grammar and improve readability. After using this tool/service, the author(s) reviewed and edited the content as needed and take(s) full responsibility for the content of the publication.

## Additional information

### Funding

| Funder | Grant reference number | Author |
| --- | --- | --- |
| National Heart Lung and Blood Institute | R01 HL146498 | Yang Zhou |
| National Institute of General Medical Sciences | P20 GM103652 | Yang Zhou |
| National Institute of General Medical Sciences | U54 GM115677 | Yang Zhou |
| National Heart Lung and Blood Institute | T32 HL134625 | Parand Sorkhdini |
| American Heart Association | 10.58275/aha. 24tpa1277918.pc.gr.196624 | Yang Zhou |
| American Thoracic Society | 23-24PHP11 | Yang Zhou |
| National Heart Lung and Blood Institute | R35 HL145242 | Michael Holtzman |
| National Institute of Allergy and Infectious Diseases | R01 AI183155 | Sanghyun Lee |

The funders had no role in study design, data collection and interpretation, or the decision to submit the work for publication.

### Author contributions

Alina Xiaoyu Yang, Conceptualization, Data curation, Software, Formal analysis, Validation, Investigation, Visualization, Methodology, Writing – original draft, Writing – review and editing; Lisa Ramos-Rodriguez, Dongqin Yang, Sonoor Majid, Methodology; Parand Sorkhdini, Carmelissa Norbrun, Conceptualization, Writing – review and editing; Sanghyun Lee, Michael Holtzman, Resources, Writing – review and editing; Yong Zhang, Resources; David F Boyd, Data curation, Formal analysis; Yang Zhou, Conceptualization, Resources, Supervision, Funding acquisition, Project administration, Writing – review and editing

### Author ORCIDs

Alina Xiaoyu Yang ⓘ https://orcid.org/0000-0003-1528-9129
Yang Zhou ⓘ https://orcid.org/0000-0001-6867-6443

### Ethics

This study was performed in strict accordance with the recommendations in the Guide for the Care and Use of Laboratory Animals of the National Institutes of Health. All of the animals were handled according to approved institutional animal care and use committee (IACUC) protocols (#24-09-0008) of Brown University. All terminal sacrifice was performed after euthanasia with urethane, and every effort was made to minimize suffering.

Reviewer #1 (Public review): https://doi.org/10.7554/eLife.104359.3.sa1
Author response https://doi.org/10.7554/eLife.104359.3.sa2

## Additional files

### Supplementary files

Supplementary file 1. PCR primer list.

Supplementary file 2. scRNA-Seq cluster annotations.

MDAR checklist

### Data availability

Sequencing data have been deposited in GEO under accession codes GSE307868.

The following dataset was generated:

| Author(s) | Year | Dataset title | Dataset URL | Database and Identifier |
|---|---|---|---|---|
| Yang AX, Ramos-Rodriguez L, Sorkhdini P, Yang D, Norbrun C, Majid S, Lee S, Zhang Y, Holtzman MJ, Boyd DF, Zhou Y | 2025 | Phospholipid Scramblase 1 (PLSCR1) Regulates Interferon-Lambda Receptor 1 (IFN-$\lambda$R1) and IFN-$\lambda$ Signaling in Influenza A Virus (IAV) Infection | https://www.ncbi.nlm.nih.gov/geo/query/acc.cgi?acc=GSE307868 | NCBI Gene Expression Omnibus, GSE307868 |

The following previously published dataset was used:

| Author(s) | Year | Dataset title | Dataset URL | Database and Identifier |
|---|---|---|---|---|
| Boyd DF, Allen EK, Randolph AG | 2020 | Spatial transcriptomics of mouse lung: Mouse D Section 1 at 10 days post influenza infection | https://www.ncbi.nlm.nih.gov/sra/?term=SRX8008853 | NCBI Sequence Read Archive, SRX8008853 |

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
