## [Editor Report · eLife Assessment]

This **valuable** manuscript presents a potentially novel mechanism by which the phospholipid scramblase, PLSCR1, defends against influenza A virus infection. The strength of the paper rests on **solid** findings involving knockout and lung specific over-expressing Plscr1 mice, airway tissue expression and mechanistic studies to show Plscr1 enhances type III interferon-mediated viral clearance.

---

## [Referee Report · Reviewer #1 (Public review)]

This manuscript by Yang et al. presents a potentially novel mechanism by which Plscr1 defends against influenza virus infection. Using a global knockout (KO) and a tissue-specific overexpression mouse model, the authors demonstrate that Plscr1-KO mice exhibit increased susceptibility and inflammation following IAV infection. In contrast, overexpression of Plscr1 in ciliated epithelial cells protects mice from infection. Through transcriptomic analysis in mice and mechanistic studies in cell culture models, the authors reveal that Plscr1 transcriptionally upregulates Ifnlr1 expression and physically interacts with this receptor on the plasma membrane, thereby enhancing IFN-λ-mediated viral clearance.

Overall, it's a well-performed study, however, causality between Plscr1 and Ifnlr1 expression needs to be more firmly established. This is because two recent studies of PLSCR1 KO cells infected with different viruses found no major differences in gene expression levels compared with their WT controls (Xu et al. Nature, 2023; LePen et al. PLoS Biol, 2024). There were also defects in the expression of other cytokines (type I and II IFNs plus TNF-alpha) so a clear explanation of why Ifnlr1 was chosen should also be given.

While Plscr1 has long been recognized as a cell-intrinsic antiviral restriction factor, few studies have explored its broader physiological role. This study thus provides interesting insights into a specific function of Plscr1 in IAV-permissive airway epithelial cells and its contribution to whole body anti-viral immunity.

Comments on revisions:

Most of the requested changes and experiments have been done. One very informative experiment is the expression of Plscr1 in Ifnlr1-KO cells to determine if it still inhibits IAV infection. The authors have indicated that this experiment is currently being pursued by crossing mice to introduce Plscr1 expression into ciliated epithelial cells on an Ifnlr1 KO background. It will show if there are Ifnlr1-independent anti-flu activities that still require Plscr1.

---

## [Author Response]

The following is the authors’ response to the original reviews

**Reviewer #1 (Public Review):**
Overall, it's a well-performed study, however, causality between Plscr1 and Ifnlr1 expression needs to be more firmly established. This is because two recent studies of PLSCR1 KO cells infected with different viruses found no major differences in gene expression levels compared with their WT controls (Xu et al. Nature, 2023; LePen et al. PLoS Biol, 2024). There were also defects in the expression of other cytokines (type I and II IFNs plus TNF-alpha) so a clear explanation of why Ifnlr1 was chosen should also be given.

We appreciate the reviewer’s reference to the two recently published research on PLSCR1’s role in SARS-CoV-2 infections. We have also discussed those studies in the Introduction and Discussion sections of this manuscript. Here, we would like to clarify ourselves for the rationale of investigating Ifn-λr1 signaling.

The reviewer mentioned “defects in the expression of other cytokines (type I and II IFNs plus TNF-alpha)” and requested a clearer explanation of why Ifnlr1 was chosen for study. In our investigation of IAV infection, we observed no defects in the expression of type I and II IFNs or TNF-α in *Plscr1-/-* mice; rather, these cytokines were expressed at even higher levels compared to WT controls (Figures 2D and 3A). This indicates that the type I and II IFN and TNF-α signaling pathways remain intact and are not negatively affected by the loss of *Plscr1*. Notably, *Ifn-λr1* expression is the only one among all IFNs and their receptors that is significantly impaired in *Plscr1-/-* mice (Figure 3A), justifying our focused investigation of this receptor. To further clarify this point, we have expanded the explanation under the section titled “Plscr1 Binds to Ifn-λr1 Promoter and Activates Ifn-λr1 Transcription in IAV Infection” within the Results. The reviewer noted that previously published studies “found no major differences in gene expression levels compared with their WT controls”, but neither study examined Ifn-λr1 expression.

(1) The authors propose that Plscr1 restricts IAV infection by regulating the type III IFN signaling pathway. While the data show a positive correlation between Ifnlr1 and Plscr1 levels in both mouse and cell culture models, additional evidence is needed to establish causality between the impaired type III IFN pathway, and the increased susceptibility observed in Plscr1-KO mice. To strengthen this conclusion, the following experiments could be undertaken: (i) Measure IAV titers in WT, Plscr1-KO, Ifnlr1-KO, and Plscr1/ Ifnlr1-double KO cells. If the antiviral activity of Plscr1 is highly dependent on Ifnlr1, there should be no further increase in IAV titers in double KO cells compared to single KO cells; (ii) over-express Plscr1 in Ifnlr1-KO cells to determine if it still inhibits IAV infection. If Plscr1's main action is to upregulate Ifnlr1, then it should not be able to rescue susceptibility since Ifnlr1 cannot be expressed in the KO background. If Plscr1 over-expression rescues viral susceptibility, then there are Ifnlr1-independent mechanisms involved. These experiments should help clarify the relative contribution of the type III IFN pathway to Plscr1-mediated antiviral immunity.

We agree with the reviewer that additional evidence is necessary to establish causality between the impaired type III IFN pathway and the increased susceptibility observed in Plscr1-KO mice. As requested by the reviewer, and one step further, we have measured IAV titers in *Wt*, *Plscr1-/-*, *Ifn-λr1-/-*, and *Plscr1-/-Ifn-λr1-/-* mouse lungs, which provided us with more comprehensive information at the tissue and organismal level compared to cell culture models. Our results are detailed under “The Anti-Influenza Activity of Plscr1 Is Highly Dependent on Ifn-λr1” within “Results” section and in Supplemental Figure 5. Importantly, there was no further increase in weight loss (Supplemental Figure 5B), total BAL cell counts (Supplemental Figure 5C), neutrophil percentages (Supplemental Figure 5D), and IAV titers (Supplemental Figure 5E) in *Plscr1-/-Ifn-λr1-/-* mouse lungs compared to *Ifn-λr1-/-* mouse lungs. These findings indicate that the antiviral activity of Plscr1 is largely dependent on Ifn-λr1.

We agree that overexpression of Plscr1 on an *Ifn-λr1-/-* background would provide additional evidence to support our conclusion from the *Plscr1-/-Ifn-λr1-/-* mice. In future studies, we plan to specifically overexpress Plscr1 in ciliated epithelial cells on the *Ifn-λr1-/-* background by breeding *Plscr1floxStopFoxj1-Cre+Ifn-λr1-/-* mice. In addition, ciliated epithelial cells isolated from *Ifn-λr1-/-* murine airways could be transduced with a *Plscr1* construct for overexpression. We hypothesize that overexpression of Plscr1 in ciliated epithelial cells will not rescue susceptibility in *Ifn-λr1-/-* mice or cells, since our *Plscr1-/-Ifn-λr1-/-* mouse model suggest that Ifn-λr1-independent anti-influenza functions of Plscr1 are likely minor compared to its role in upregulating Ifn-λr1. These future plans have been added to the “Discussion” section, and we look forward to presenting our results in a forthcoming publication.

(3) In Figure 4, the authors demonstrate the interaction between Plscr1 and Ifnlr1. They suggest that this interaction modulates IFN-λ signaling. However, Figures 5C-E show that the 5CA mutant, which lacks surface localization and the ability to bind Ifnlr1, exhibits similar anti-flu activity to WT Plscr1. Does this mean the interaction between Plscr1 and Ifnlr1 is dispensable for Plscr1-mediated antiviral function? Can the authors compare the activation of IFN-λ signaling pathway in Plscr1-KO cells expressing empty vector, WT Plscr1, and 5CA mutant? This could be done by measuring downstream ISG expression or using an ISRE-luciferase reporter assay upon IFN-λ treatment.

We agree with the reviewer that downstream activation of the IFN-λ signaling pathway is a critical component of the proposed regulatory role of PLSCR1. As suggested, we attempted to perform an ISRE-luciferase reporter assay following IFN-λ treatment in PLSCR1 rescue cell lines by transfecting the cells with hGAPDH-rLuc (Addgene #82479) and pGL4.45 [luc2P/ISRE/Hygro] (Promega #E4041).

Despite extensive efforts over several months, we were unable to achieve expression of pGL4.45 [luc2P/ISRE/Hygro] in PLSCR1 rescue cells using either Lipofectamine 3000 or electroporation, as no firefly luciferase activity was detected at baseline or following IFN-λ treatment. In contrast, hGAPDH-rLuc was robustly expressed in these cells.

The pGL4.45 [luc2P/ISRE/Hygro] plasmid was obtained directly from Promega as a purified product, and its sequence was confirmed via whole plasmid sequencing. Additionally, both hGAPDH-rLuc and pGL4.45 [luc2P/ISRE/Hygro] were successfully expressed in 293T cells, indicating that neither the plasmids nor the transfection protocols are inherently faulty.

We suspect that prior modifications to the PLSCR1 rescue cells—such as CRISPR-mediated knockout and lentiviral transduction—may interfere with successful transfection of pGL4.45 [luc2P/ISRE/Hygro] through an as-yet-unknown mechanism. Although these results are disappointing, we will continue troubleshooting and plan to communicate in a separate manuscript once the luciferase assay is successfully established.

**Reviewer #1 (Recommendations):**
(1) In the introduction, the linkage between the paragraph discussing type III IFN and PLSCR1 needs to be better established. The mention of PLSCR1 being an ISG at the outset may help connect these two paragraphs and make the text appear more logical.

We apologize for the lack of linkage and logic between type 3 IFN and PLSCR1. We have introduced PLSCR1 as an ISG at the beginning of its paragraph as recommended.

(2) The statement that, “Intriguingly, PLSCR1 is also an antiviral ISG, as its expression can be highly induced by type 1 and 2 interferons in various viral infections[15, 16]. However, whether its expression can be similarly induced by type 3 interferon has not been studied yet.” is incorrect. Xu et al. tested the role of PLSCR1 in type III IFN-induced control of SARS-CoV-2 (ref. 24). This needs to be revised.

We apologize for the incorrect information in the introduction and have revised the paragraph with the proper citation.

(3) In Figure 3B, can the authors provide a comprehensive heatmap that includes all ISGs above the threshold, rather than only a subset? This would offer a more complete overview of the changes in type I, II, and III IFN pathways in Plscr1-KO mice.

As suggested by the reviewer, we have provided a comprehensive heatmap that includes all ISGs above the threshold in Figure 3C (previously Figure 3B). We identified a total of 1,113 ISGs in our dataset with a fold change ≥2. Enlarged heatmaps with gene names are provided in Supplemental Figure 1. Among those ISGs, 584 are regulated exclusively by type 1 IFNs, and 488 are regulated by both type 1 and type 2 interferons. Unfortunately, the Interferome database does not include information on type 3 IFN-inducible genes in mice[1]. Although many ISGs were robustly upregulated in *Plscr1-/-* infected lungs, consistent with inflammation data, a large subset of ISGs failed to be transcribed when *Ifn-λr1* function was impaired, especially at 7 dpi. We suspect that those non-transcribed ISGs in *Plscr1-/-* mice may be specifically regulated by type 3 IFN and represent interesting targets for future research. These results have been added to “Plscr1 Binds to *Ifn-λr1* Promoter and Activates *Ifn-λr1* Transcription in IAV Infection” within “Results” section.

(4) In Figure 3C, 5B and 7H, immunoblots should also be included to measure changes of Ifnlr1/IFNLR1 protein level.

As requested by the reviewer, we have provided western blots measuring Ifn-λr1/IFN-λR1 protein level in Figure 5B and 7I. The protein expressions were consistent with the PCR results.

(5) In Figure 3H, the amount of RPL30 is also low in the anti-PLSCR1-treated and IgG samples, making it difficult to estimate if ChIP binding is genuinely impacted.

RPL30 Exon 3 serves as a negative control in the ChIP experiment and is not expected to bind either the anti-PLSCR1-treated or the IgG control samples. Anti-Histone H3 treatment is a positive control, with the treated sample expected to show binding to RPL30 Exon 3. We hope this clarification has addressed any further potential confusion from the reviewer.

(6) In Figure 4A, can the authors show a larger slice of the gel with molecular weight markers for both Plscr1 and Ifnlr1. In the coIP, the binding may be indirect through intermediate partners. Proximity ligation assay is a more direct assay for interaction and can be stated as such.

As suggested by the reviewer, we have included whole gel images of Figure 4A with molecular weight markers for both Plscr1 and Ifnlr1 in Supplemental Figure 3. We appreciate the reviewer’s affirmation of proximity ligation assay and have stated it as a more direct assay for interaction under “Plscr1 Interacts with Ifn-λr1 on Pulmonary Epithelial Cell Membrane in IAV Infection” in “Results” section.

(7) In Figure 5A, how is the expression of PLSCR1 WT and mutants driven by an EF-1α promoter can be further upregulated by IAV infection? Can the authors also use immunoblots to examine the protein level of PLSCR1?

We apologize for the confusion and appreciate the reviewer’s careful observation. We were initially surprised by this finding as well, but upon further investigation, we found out that the human *PLSCR1* primers used in our qRT-PCR assay can still detect the transcription from the undisturbed portion of the endogenous *PLSCR1* mRNA, even in *PLSCR1-/-* cells. In the original Figure 5A, data for vector-transduced PLSCR1^-/-^ were not included because PCR was not performed on those samples at the time. After conducting PCR for vector-transduced PLSCR1^-/-^ cells, we detected transcription of *PLSCR1*, which confirms that the signaling originates from endogenous DNA, but not from the EF-1α promoter-driven *PLSCR1* plasmid. Please see Author response image 1 below.

**Author response image 1. sa2fig1:** 

The forward human *PLSCR1* primer we used matches 15-34 nt of *Wt PLSCR1*, and the reverse primer matches 224-244 nt of *Wt PLSCR1*. CRISPR-Cas9 KO of *PLSCR1* was mediated by sgRNAs in A549 cells and was performed by Xu et al[2]. sgRNA #1 matches 227-246 nt, sgRNA #2 matches 209-228 nt, and sgRNA #3 matches 689-708 nt of *Wt PLSCR1.* The sgRNAs likely introduced a short deletion or insertion that does not affect transcription. However, those endogenous mRNA transcripts cannot be translated to functional and detectable PLSCR1 proteins, as validated by our western blot (below), as well as western blots performed by Xu et al[2]. Therefore, our primers could amplify endogenous *PLSCR1* transcripts upregulated by IAV infection, if 15-244 nt was not disturbed by CRISPR-Cas9 KO. By western blot, we confirmed that only endogenous PLSCR1 expression is upregulated by IAV infection, and exogenous protein expression of PLSCR1 plasmids driven by an EF-1α promoter are not upregulated by IAV infection.

To avoid confusion, we have removed the original Figure 5A from the manuscript.

(8) In Figure 5C, the loss of anti-flu activity with the H262Y mutant is modest, suggesting the loss of ifnlr1 transcription is only partly responsible for the susceptibility of Plscr1 KO cells. The anti-flu activity being independent of scramblase activity resembles the earlier discovery of SARS-CoV-2 (Xu et al., 2024). This could be stated in the results since it is an important point that scramblase activity is dispensable for several major human viruses and shifts the emphasis regarding mechanism. It has been appropriately noted in the discussion.

We appreciated the comments and have acknowledged the consistency of our results with those of Xu et al. under “Both Cell Surface and Nuclear PLSCR1 Regulates IFN-λ Signaling and Limits IAV Infection Independent of Its Enzymatic Activity” in the “Results” section.

**Reviewer #2 (Recommendations):**
(1) The statement that type I interferons are expressed by “almost all cells” is inaccurate (line 61). Type I IFN production is also context-dependent and often restricted to specific cell types upon infection or stimulation.

We apologize for the inaccurate description of the expression pattern of type 1 IFNs and have corrected the restricted cellular sources of type 1 IFNs in the “Introduction”.

(2) The antiviral response is assessed solely through flu M gene expression. Incorporating infectious virus titers (e.g., TCID50 or plaque assay) would provide a more robust and direct measure of antiviral activity.

As requested by the reviewer, we have performed plaque assays on all experiments where flu M gene expression levels were measured (Figure 1G, 5E and 7F, and Supplemental Figure 6E). The plaque assay results are consistent with the flu M gene expressions.

(3) While mRNA expression of interferons is measured, protein levels (e.g., through ELISA) should also be quantified to establish the functional relevance of IFN expression changes.

As requested by the reviewer, we have quantified the protein level of IFN-λ in mouse BAL with ELISA (Figure 2E). The ELISA results are consistent with the mRNA expressions of IFN-λ.

(4) It is unclear whether reduced IFNLR1 expression translates to defective downstream signaling or antiviral responses after IFN-λ treatment in PLSCR1-deficient cells. This is particularly pertinent given the increase in IFN-λ ligand in vivo, which might compensate for receptor downregulation.

We agree with the reviewer that downstream activation of the IFN-λ signaling pathway is a critical aspect of PLSCR1’s proposed regulatory role. To investigate this, we attempted an ISRE-luciferase reporter assay to assess downstream signaling following IFN-λ treatment in PLSCR1 rescue cells. Unfortunately, the experiment encountered unforeseen technical issues. For additional context, please refer to our response to Reviewer #1’s public review #3.

(5) Detailed gating strategies for immune cell subsets are absent and should be included for clarity and reproducibility.

We would like to clarify that the immune cell subsets in BAL fluids were counted manually following cytospin preparation and Diff-Quik staining (Figure 2B and 7H, and Supplemental Figures 2C, 5D, and 8D), rather than by flow cytometry. We hope this resolves the reviewer’s confusion.

(6) The study does not definitively establish that reduced IFN-λ signaling causes the observed in vivo phenotype. Increased morbidity and mortality in PLSCR1-deficient mice could also stem from elevated TNF-α levels and lung damage, as proinflammatory cytokines and/or enhanced lung damage are known contributors to influenza morbidity and mortality. This point warrants detailed discussions.

We agreed with the reviewer that this study does not guarantee a definitive causality between reduced IFN-λ signaling and increased morbidity of *Plscr1-/-* mice and more experiments are needed to reach the conclusion. We have acknowledged this limitation of our study in the “Discussion”, as requested by the reviewer. We hope to fully eliminate the confounding elements and definitively establish the proposed causality in future studies.

**Reviewer #3 (Public review):**
Summary:Yang et al. have investigated the role of PLSCR1, an antiviral interferon-stimulated gene (ISG), in host protection against IAV infection. Although some antiviral effects of PLSCR1 have been described, its full activity remains incompletely understood.This study now shows that Plscr1 expression is induced by IAV infection in the respiratory epithelium, and Plscr1 acts to increase Ifn-λr1 expression and enhance IFN-λ signaling possibly through protein-protein interactions on the cell membrane.Strengths:The study sheds light on the way Ifnlr1 expression is regulated, an area of research where little is known. The study is extensive and well-performed with relevant genetically modified mouse models and tools.Weaknesses:There are some issues that need to be clarified/corrected in the results and figures as presented.Also, the study does not provide much information about the role of PLSCR1 in the regulation of Ifn-λr1 expression and function in immune cells. This would have been a plus.

We would like to thank the reviewer for the positive feedback and insightful comment regarding the roles of PLSCR1 and IFN-λR1 in immune cells. It is important to note that IFN-λR1 expression is highly restricted in immune cells and is primarily limited to neutrophils and dendritic cells[3]. While dendritic cells were not the focus of this study, we did examine all immune cell subsets in our single cell RNA seq data and performed infection experiments in *Plscr1floxStop/LysM-Cre+* mice. We have not observed any significant findings in these populations. On the other hand, we do have some interesting preliminary data suggesting a role for PLSCR1 in regulating Ifn-λr1 expression and function in neutrophils. These findings are discussed in detail in our response to reviewer #3’s recommendation #12.

**Reviewer #3 (Recommendations):**
(1) In Figure 1B, the Plscr1 label should be moved to the y-axis so that readers don't confuse it with the Plscr1-/- mice used in the other figure panels. The fact that WT mice were used should be added in the figure legend.

We apologize for the confusion in the figures. We have moved Plscr1 label to the y-axis in Figure 1B and have mentioned *Wt* mice were used in the figure legend.

(2) In Figure 1C and D, the type of dose leading to the presented data should be added to help the reader. Also, shouldn't statistics be added?

We appreciate the suggestion and have added doses to Figure 1C and 1D. We are confused about the request of adding statistics by the reviewer, as two-way ANOVA tests were used to compare weight losses, and the significance was labeled on the figures.

(3) In Figures 1, F, and G, it is not indicated whether sublethal or lethal dose was used for the IAV infection. This should be very clear in the figure and figure legend.

We apologize for the confusion of infection doses used in the figures. We have added doses to Figure 1F, 1G and 1H.

(4) In Figure 1, the CTCF abbreviation should be explained in the Figure legend.

We have explained CTCF in the figure legend as requested.

(5) In Figure 2B, this is percentages of what?

Figure 2B shows the percentages of each immune cell type within total BAL cells.

(6) In Figures 3A and B, transcriptomes for each condition are from how many mice? Also, what do heatmaps show? Fold induction, differences, etc, and from what? What is compared with what? In addition, is there a discordance between the RNAseq data of Figure 3A and the qPCR data of Fig. 3C in terms of Ifnlr1 expression?

In Figure 3A and 3C (previously 3B), RNA from the whole lungs of 9 mice per PBS-treated group and 4 mice per IAV-infected group were pooled for transcriptomic analysis. Figure 3A represents a heatmap of differential gene expression, while Figure 3C (previously 3B) represents fold changes in gene expression relative to uninfected controls. In both heatmaps, gene expression values are color-coded from row minimum (blue) to row maximum (red), enabling comparison across groups within each gene (row). The major comparison of interest in these heatmaps is between *Wt* infected mice versus *Plscr1-/-* infected mice. We have added this information to the figure legend.

We also acknowledge the reviewer’s observation regarding the discordance between the RNA seq data of Figure 3A and the qPCR data of Figure 3B (previously 3C) for Ifnlr1 expression. To address this, we have repeated the qRT-PCR experiment with additional samples at 7 dpi. In the updated results, *Wt* mice consistently show significantly higher *Ifn-λr1* expression than *Plscr1-/-* infected mice at both 3 dpi and 7 dpi, consistent with the RNA seq data. However, a time-dependent discrepancy between the RNA-seq and qRT-PCR datasets remains: *Ifn-λr1* expression continues to increase at 7 dpi in the RNA-seq data (Figure 3A), whereas it declines in the qRT-PCR results (Figure 3B). The reason for this discrepancy remains unclear and has been addressed in the Discussion section.

(7) In Figure 3D, have the authors checked whether the Ifnlr1 antibody they use is indeed specific for Ifnlr1? Have they used any blocking peptide for the anti-mouse Ifn-λr1 polyclonal antibody they are using? Also, in Figure 3E, the marker used for staining should be indicated in the pictures of the lung section.

Unfortunately, a blocking peptide is not available for the anti-mouse Ifn-λr1 polyclonal antibody used in our study. To assess antibody specificity, we have performed immunofluorescence staining of Ifn-λr1 on lung tissues from *Ifn-λr1-/-* mice using the same antibody. No signal was detected (Supplemental Figure 5A), supporting the specificity of the antibody for Ifn-λr1.

As requested by the reviewer, we have added the marker (Ifn-λr1) to the pictures of the lung section in Figure 3E.

(8) In Figure 5, it's better to move each graph's label that stands to the top (e.g. PLSCR1, IFN-λR1 etc) to the y-axis label so that it doesn't get confused with the mouse -/- label.

We apologize for the confusion and have moved the top label to the y-axis in Figure 5.

(9) In Figure 6A, it is claimed that the 'two-dimensional UMAP demonstrated that these main lung cell populations (epithelial, endothelial, mesenchymal, and immune) were dynamic over the course of infection.'. This is not clear by the data. The percentage of cells per cluster should be calculated.

As requested by the reviewer, the proportion (Supplemental Figure 6A) and cell count (Supplemental Figure 6B) of each cluster have been calculated and included in “PLSCR1 Expression Is Upregulated in the Ciliated Airway Epithelial Compartment of Mice following Flu Infection” under “Results” section. Together with the two-dimensional UMAP (Figure 6A), these data demonstrate that the main lung cell populations (epithelial, endothelial, mesenchymal, and immune) were dynamic over the course of infection. Following infection, many populations emerged, particularly within the immune cell clusters. At the same time, some clusters were initially depleted and later restored, such as microvascular endothelial cells (cluster 2). Other populations, such as interferon-responsive fibroblasts (cluster 20), showed a dramatic yet transient expansion during acute infection and disappeared after infection resolved.

(10) In Figure 6 B and C, the legend should indicate that these are Violin plots. Also, if AT2 cells don't express Plscr1, does that indicate that in these cells Plscr1 is not needed for IFN-λR1 expression?

As requested, we have indicated in the legend of Figure 6B and 6C that these are violin plots. *Plscr1* is expressed at low levels in AT2 cells. However, it is unclear whether Plscr1 is needed for Ifn-λr1 expression in AT2 cells, and it would be interesting to investigate further.

(11) In lines 302-304, it is stated that 'Among the various epithelial populations, ciliated epithelial cells not only had 303 the highest aggregated expression of Plscr1, but also were the only epithelial cell 304 population in which significantly more Plscr1 was induced in response to IAV infection.'. Which data/ figure support this statement?

Figure 6B shows that among the various epithelial populations, ciliated epithelial cells had the highest aggregated expression of *Plscr1*. To better illustrate this statement, we have rearranged the order of cell clusters from highest to lowest *Plscr1* expression, and added red dots to indicate the mean expression levels for each cluster in Figure 6B.

Ciliated epithelial cells also had the most significant increase in *Plscr1* expression (p < 2.22e-16 and p = 6.7e-05) in early IAV infection at 3 dpi (Figure 6C and Supplemental Figure 7A-7K). In comparison, AT1 cells were the only other epithelial cluster to show *Plscr1* upregulation at 3dpi, but to a much less extent (p = 0.033, Supplemental Figure 7J). Supplemental Figure 7 was added to better support the statement and the explanation was added to “PLSCR1 Expression Is Upregulated in the Ciliated Airway Epithelial Compartment of Mice following Flu Infection” under “Results” section.

(12) As earlier, if Plscr1 is not expressed in neutrophils (Figure 6F), does that mean IFN-λR1 expression does not require Plscr1 in these cells?

Although *Plscr1* is expressed at lower levels in neutrophils compared to epithelial cells, it is still detectable. In fact, our preliminary data suggest that *IFN-λR1* expression in neutrophils is dependent on *Plscr1*. We have isolated neutrophils from peripheral blood and BAL of IAV-infected *Wt* and *Plscr1-/-* mice using a mouse neutrophil enrichment kit. Quantitative PCR results showed that *Plscr1-/-* neutrophils exhibit significantly lower expression of *Ifn-λr1*, alongside elevated levels of *Il-1β*, *Il-6* and *Tnf-α* in IAV infection (see figures below). These findings suggest that Plscr1 may play an anti-inflammatory role in neutrophils by upregulating Ifn-λr1. These data were not included in the current manuscript because they are beyond the scope of current study, but we hope to address the role of PLSCR1 in regulating *IFN-λR1* expression and function in neutrophils in a future study.

**Author response image 3. sa2fig3:** 

(13) The Figure 7A legend is not well stated. Something like ' Schematic representation of the experimental design of...' should be included. Also, Figure 7J is not referenced in the text.

We apologize for the unclear Figure 7A legend and have changed it to “Schematic representation of the experimental design of ciliated epithelial cell conditional *Plscr1 KI* mice.” Figure 8 (previously Figure 7J) has now been referenced in the text.

(14) In the Methods, more specific information in some parts should be provided. For example, the clones of the antibodies used should be included.Apart from the 10x technology, the kits used and the type of the Illumina sequencing should be provided. Information on how the QC was performed (threshold for reads/cell, detected genes/per cells, and % of mitochondrial genes etc) should be added.

We apologize for the missing information in the “Methods”. We have now provided the clones of the antibodies used, the kit used to generate single-cell transcriptomic libraries, the type of the Illumina sequencing, and the QC performance data.

References

(1) Rusinova, I., et al., Interferome v2.0: an updated database of annotated interferon-regulated genes. Nucleic Acids Res, 2013. 41(Database issue): p. D1040-6.

(2) Xu, D., et al., PLSCR1 is a cell-autonomous defence factor against SARS-CoV-2 infection. Nature, 2023. 619(7971): p. 819-827.

(3) Donnelly, R.P., et al., The expanded family of class II cytokines that share the IL-10 receptor-2 (IL-10R2) chain. J Leukoc Biol, 2004. 76(2): p. 314-21.